# RAG suppresses group 2 innate lymphoid cells

Aaron M Ver Heul[1], Madison Mack[2], Lydia Zamidar[3,4,5,6], Masato Tamari[3,4,5,6], Ting-Lin Yang[7], Anna M Trier[7], Do-Hyun Kim[8,9], Hannah Janzen-Meza[1], Steven J Van Dyken[8], Chyi-Song Hsieh[10], Jenny M Karo[11,12], Joseph C Sun[11,12], Brian S Kim[3,4,5,6,13]*

[1]Division of Allergy and Immunology, Department of Medicine, Washington University School of Medicine, St. Louis, United States; [2]Immunology and Inflammation Research Therapeutic Area, Sanofi, Cambridge, United States; [3]Kimberly and Eric J. Waldman Department of Dermatology, Icahn School of Medicine at Mount Sinai, New York, United States; [4]Mark Lebwohl Center for Neuroinflammation and Sensation, Icahn School of Medicine at Mount Sinai, New York, United States; [5]Marc and Jennifer Lipschultz Precision Immunology Institute, Icahn School of Medicine at Mount Sinai, New York, United States; [6]Friedman Brain Institute, Icahn School of Medicine at Mount Sinai, New York, United States; [7]Division of Dermatology, Department of Medicine, Washington University School of Medicine, St. Louis, United States; [8]Department of Pathology and Immunology, Washington University School of Medicine, St. Louis, United States; [9]Department of Life Science, College of Natural Sciences, Hanyang University, Seoul, Republic of Korea; [10]Division of Rheumatology, Department of Medicine, Washington University School of Medicine, St. Louis, United States; [11]Immunology and Microbial Pathogenesis Program, Graduate School of Medical Sciences, Weill Cornell Medical College, New York, United States; [12]Immunology Program, Memorial Sloan Kettering Cancer Center, New York, United States; [13]Allen Discovery Center for Neuroimmune Interactions, Icahn School of Medicine at Mount Sinai, New York, United States

*For correspondence:
brian.kim3@mountsinai.org

## eLife Assessment

This study provides new insights into the expression profile of ILCs that demonstrate a history of RAG expression. It examines in part the potential intrinsic regulation of RAG expression and seeks to understand how the epigenetic state of ILCs is established, although a full understanding of intrinsic factors is only partially supported. The work provides a **convincing** and **important** molecular dataset, and strengthens our understanding of intrinsic regulation, and would be of interest more broadly to cell biologists seeking to understand immune cell development.

**Abstract** Antigen specificity is the central trait distinguishing adaptive from innate immune function. Assembly of antigen-specific T cell and B cell receptors occurs through V(D)J recombination mediated by the Recombinase Activating Gene endonucleases RAG1 and RAG2 (collectively called RAG). In the absence of RAG, mature T and B cells do not develop and thus RAG is critically associated with adaptive immune function. In addition to adaptive T helper 2 (Th2) cells, group 2 innate lymphoid cells (ILC2s) contribute to type 2 immune responses by producing cytokines like Interleukin-5 (IL-5) and IL-13. Although it has been reported that RAG expression modulates the function of innate natural killer (NK) cells, whether other innate immune cells such as ILC2s are

affected by RAG remains unclear. We find that in RAG-deficient mice, ILC2 populations expand and produce increased IL-5 and IL-13 at steady state and contribute to increased inflammation in atopic dermatitis (AD)-like disease. Furthermore, we show that RAG modulates ILC2 function in a cell-intrinsic manner independent of the absence or presence of adaptive T and B lymphocytes. Lastly, employing multiomic single cell analyses of RAG1 lineage-traced cells, we identify key transcriptional and epigenomic ILC2 functional programs that are suppressed by a history of RAG expression. Collectively, our data reveal a novel role for RAG in modulating innate type 2 immunity through suppression of ILC2s.

## Introduction

Atopic disorders such as AD, asthma, and food allergy are associated with Th2 cell responses, elevated production of the type 2 cytokines IL-4, IL-5, and IL-13, and induction of immunoglobulin(Ig) E (*Ständer, 2021*; *Hammad and Lambrecht, 2021*; *Yu et al., 2016*; *Tordesillas et al., 2017*). Classically, this allergic inflammatory cascade is believed to originate with antigenic stimulation of T cell receptors on adaptive T cells, which in turn results in the production of IgE from B and plasma cells capable of binding the same antigen. Indeed, the presence of antigen-specific IgE reactivity is a hallmark of atopic disorders (*Ring, 2014*). Thus, for decades, antigen-specific adaptive Th2 cell responses have been the primary focus of investigation in the pathogenesis of atopic diseases. However, recent studies indicate that innate immune cells are sufficient to not only drive allergic pathology, but also amplify adaptive Th2 cell responses (*Halim et al., 2016*; *Halim et al., 2014*; *Sokol et al., 2009*; *Perrigoue et al., 2009*). These studies suggest that innate immune mechanisms may play a larger role in driving atopic inflammation than previously recognized.

ILCs, while lacking antigen receptors generated by RAG activity, are the innate counterparts of T cells. For example, ILC2s mirror adaptive Th2 cells in their developmental requirements, cytokine profiles, and effector functions (*Vivier et al., 2018*). Unlike classical T cells, ILC2s are concentrated at barrier surfaces to rapidly respond to microbial and environmental stimuli and are key mediators of inflammatory skin conditions like AD (*Imai et al., 2013*; *Salimi et al., 2013*; *Kim et al., 2013*). Indeed, in murine models of AD-like disease, type 2 skin inflammation can still occur despite the absence of adaptive T cells, but is further reduced after depletion of ILC2s (*Salimi et al., 2013*; *Kim et al., 2013*). Furthermore, recent studies have shown that ILC2s harbor non-redundant functions for anti-helminth immunity alongside the adaptive immune system (*Tsou et al., 2022*; *Jarick et al., 2022*). These findings suggest that ILC2 dysfunction may also uniquely contribute to the pathogenesis of atopic diseases, independent of adaptive immunity. However, the cell-intrinsic mechanisms that drive ILC2 dysregulation remain poorly understood.

ILC2s were originally discovered due to their capacity to orchestrate multiple allergic pathologies in immunocompromised mice, most notably in RAG-deficient mice that lack T and B cells (*Fort et al., 2001*; *Hurst et al., 2002*; *Fallon et al., 2006*; *Moro et al., 2010*; *Saenz et al., 2010*; *Neill et al., 2010*; *Price et al., 2010*). These discoveries fundamentally redefined our understanding of allergic diseases and placed a major focus on ILC2s as potential drivers of human allergic disease. However, despite ILC2s not requiring RAG expression for their development, fate mapping studies in mice have demonstrated that up to 60% of ILC2s have historically expressed RAG1 during development (*Yang et al., 2011*; *Karo et al., 2014*). Although previous work has described the roles of RAG beyond antigen receptor recombination in developing T and B cells (*Bredemeyer et al., 2008*; *Bednarski et al., 2012*; *Teng et al., 2015*) and NK cells (*Karo et al., 2014*), how this developmental expression of RAG impacts ILC2s remains unclear.

By directly comparing RAG-deficient and RAG-sufficient mice, we unexpectedly found enhanced AD-like disease in RAG-deficient mice, despite the lack of adaptive lymphocytes predicted to underlie AD-like inflammation. Using splenocyte replenishment and bone marrow chimeras, we show that RAG suppresses ILC2 activation and expansion in a cell-intrinsic manner. Employing a RAG1-lineage reporter mouse line, we performed simultaneous single-cell multiomic RNA and ATAC sequencing to show that RAG fate-mapped ILC2s display unique transcriptional and epigenomic alterations consistent with the suppression of effector cytokine production. Collectively, our studies reveal evolutionarily conserved regulatory functions of RAG within innate lymphocytes, extending beyond the generation of antigen receptors in adaptive lymphocytes.

## Results

### RAG deficiency leads to the expansion and activation of ILC2s

AD-like disease can be elicited in the skin of mice with repeated application of the topical vitamin D analog calcipotriol (MC903) (*Li et al., 2006*). Although it has been previously demonstrated that MC903 can induce AD-like disease in RAG-deficient mice that lack T and B cells, in part via ILC2 activation (*Salimi et al., 2013*; *Kim et al., 2013*), the relative contributions of ILC2s and the adaptive lymphocyte compartment have not been rigorously evaluated. We hypothesized that the presence of Th2 cells, in addition to ILC2s, would lead to enhanced AD-like disease in an additive fashion. In testing this, we evaluated both RAG1-sufficient wild-type (WT) mice and RAG1-deficient *Rag1⁻/⁻* mice in the setting of AD-like disease (*Figure 1A*). Unexpectedly, we observed that *Rag1⁻/⁻* mice developed increased ear skin thickness (*Figure 1B*) and increased absolute numbers and proportion of ILC2s in the skin-draining lymph nodes (sdLNs) compared to control WT mice (*Figure 1C and D*; *Figure 1—figure supplement 1A and B*). Furthermore, a larger proportion of ILC2s from *Rag1⁻/⁻* mice exhibited production of both IL-5 (*Figure 1E*; *Figure 1—figure supplement 1C*) and IL-13 (*Figure 1F*; *Figure 1—figure supplement 1C*) following ex vivo stimulation and intracellular cytokine staining. Our findings indicated that RAG1 deficiency results in paradoxically worse AD-like disease in association with enhanced ILC2 expansion and activation.

To determine whether this phenomenon was specific to AD-like pathological conditions, we next examined the sdLNs in *Rag1⁻/⁻* and lymphocyte-sufficient *Rag1⁺/⁻* littermate control mice in the absence of disease (*Figure 1G*). We found that the absolute number and frequency of ILC2s was increased at steady state in *Rag1⁻/⁻* sdLNs (*Figure 1H and I*) and that a higher proportion of these ILC2s produced both IL-5 (*Figure 1J*) and IL-13 (*Figure 1K*) compared to WT controls. The RAG recombinase requires both RAG1 and RAG2 components to successfully rearrange a functional antigen receptor in adaptive lymphocytes (*Liu et al., 2022*). Thus, to test whether our findings are specific to RAG1, or related to the function of the overall RAG complex, we similarly examined the steady-state profile of ILC2s in *Rag2⁻/⁻* mice (*Figure 1—figure supplement 2A*). Deficiency of RAG2 led to an expansion of ILC2s in the sdLNs (*Figure 1—figure supplement 2B and C*) and increased proportions of ILC2s expressing IL-5 (*Figure 1—figure supplement 2D*) and IL-13 (*Figure 1—figure supplement 2E*) similar to *Rag1⁻/⁻* mice. Collectively, these findings suggest that the RAG recombinase modulates ILC2 function at steady state and during type 2 inflammation. However, whether the hyperactive ILC2 phenotype is due to a cell-intrinsic process or simply due to the absence of T and B cells was unclear.

### ILC2 suppression by RAG is cell intrinsic

Given the importance of the adaptive lymphocyte compartment in shaping the secondary lymphoid organ repertoire, we next wanted to examine whether the presence of adaptive lymphocytes could restore ILC2 homeostasis in RAG-deficient mice. To test this, we created splenocyte chimera mice by reconstituting both *Rag1⁻/⁻* and control WT mice with splenocytes containing T and B cells from WT donor mice (*Figure 2A*). We first assessed the overall level of immune reconstitution in the recipient mice and found fully restored proportions of CD4⁺ (*Figure 2—figure supplement 1A*) and CD8⁺ (*Figure 2—figure supplement 1B*) T cells in the spleens of recipient *Rag1⁻/⁻* mice, although B cell numbers remained significantly lower than in WT mice (*Figure 2—figure supplement 1C*). Upon induction of AD-like disease, we found that the *Rag1⁻/⁻* mice still exhibited increased ear skin thickness (*Figure 2B*), enhanced expansion of ILC2s (*Figure 2C and D*), and increased proportions of ILC2s expressing IL-5 (*Figure 2E*) and IL-13 (*Figure 2F*) in the sdLNs. Interestingly, we found significantly higher proportions of eosinophils in the spleens of *Rag1⁻/⁻* recipient mice (*Figure 2—figure supplement 1D*), possibly reflecting the increased IL-5 production we observed from ILC2s. These findings indicate that the mere introduction of exogenous T and B cells is not sufficient to suppress ILC2 dysregulation in the setting of RAG deficiency.

To further test whether this phenotype is mediated by cell-intrinsic RAG expression, we next generated mixed bone marrow (BM) chimeras. We harvested BM from congenic CD90.1⁺ WT and CD90.2⁺ *Rag1⁻/⁻* donor mice on a CD45.2⁺ background and reconstituted sub-lethally irradiated CD45.1⁺ congenic WT recipients with a 50:50 mixture of WT:*Rag1⁻/⁻* BM (*Figure 2G*). After confirming the reconstitution of donor immune cells in the sdLN (*Figure 2—figure supplement 2A and D*), we examined the frequency and activity of ILC2s in the sdLNs based on whether they originated from WT (CD90.1⁺) or *Rag1⁻/⁻* (CD90.2⁺) donors (*Figure 2—figure supplement 2B and C*). Strikingly, of the total

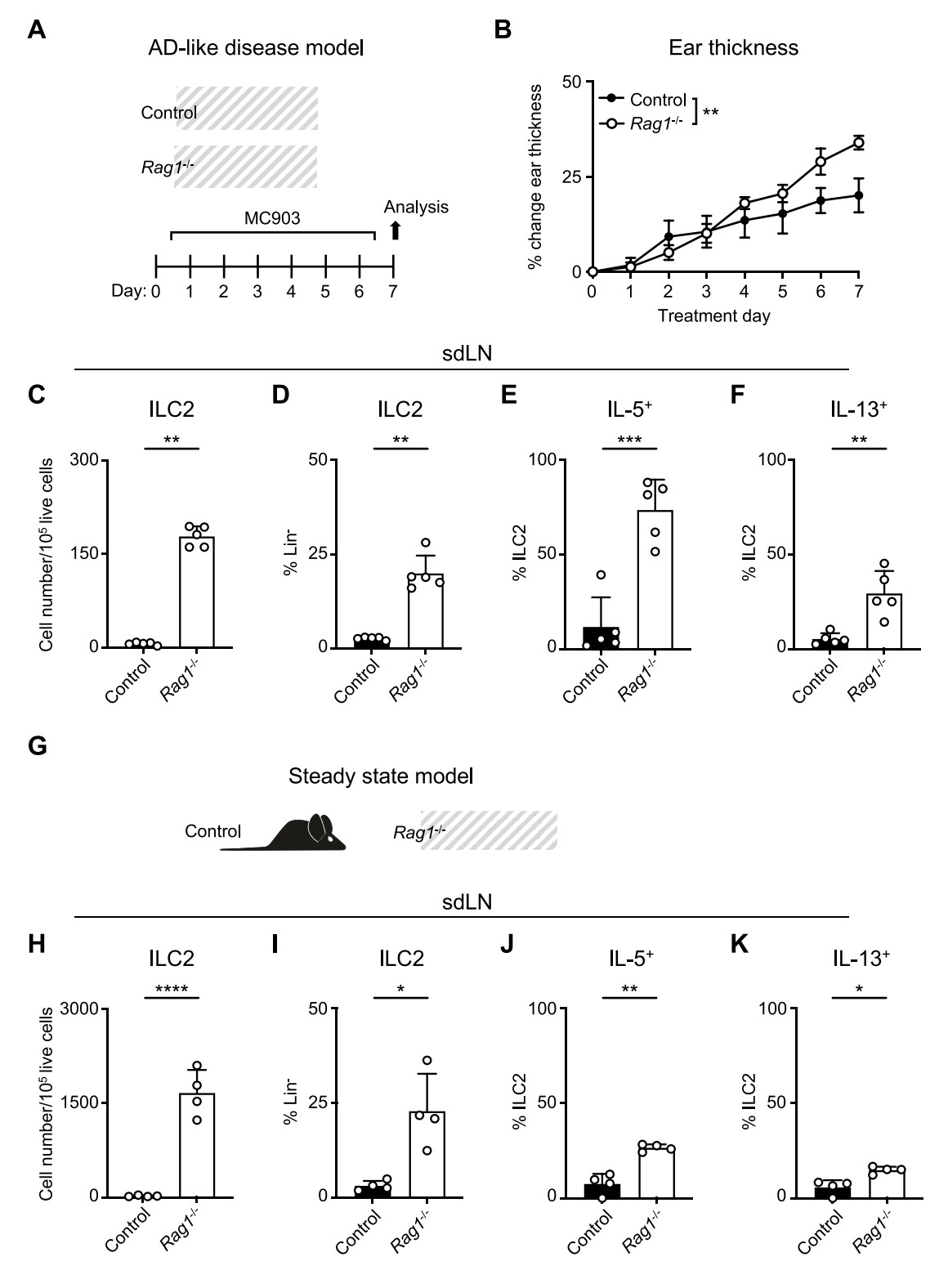

**Figure 1.** Recombinase activating gene (RAG) deficiency leads to the expansion and activation of group 2 innate lymphoid cells (ILC2s) during inflammation and at a steady state. (**A**) Experimental schematic of atopic dermatitis (AD)-like disease. Wild-type (WT) B6 (Control) mice or *Rag1*^-/- mice treated topically to the inner surface of each ear with 2 nmol MC903 in 10 µL ethanol vehicle daily for 7 days develop AD-like inflammation. (**B**) Ear thickness measured daily in AD-like inflammation. Data representative of at least two independent experiments, 5 mice/group. ** p<0.01 by two-way

*Figure 1 continued on next page*

*Figure 1 continued*

ANOVA with Sidak's multiple comparisons test, day 7. (**C**) Total number ILC2s normalized to $10^5$ live cells and (**D**) proportion of CD90$^+$, Lin$^-$ cells (Lin$^-$ defined as CD3$^-$, CD5$^-$, CD11b$^-$, CD11c$^-$, CD19$^-$, NK1.1$^-$, and FcεR1$^-$) determined to be ILC2s (IL-33R$^+$) in skin-draining lymph nodes (sdLN) from WT or *Rag1*$^{-/-}$ mice with AD-like ear inflammation. Percent ILC2 from sdLN in AD-like disease following Phorbol 12-myristate 13-acetate (PMA)/ionomycin stimulation positive for (**E**) IL-5 or (**F**) IL-13 staining. (**G**) Schematic of steady state analysis of sdLN from WT (Control) or *Rag1*$^{-/-}$ mice. (**H**) Total number ILC2s normalized to $10^5$ live cells and (**I**) ILC2 proportion of steady state sdLN CD90$^+$, Lin$^-$ cells determined to be ILC2s as in (**C, D**). Percent ILC2 from sdLN in steady state following PMA/ionomycin stimulation positive for (**J**) IL-5 or (**K**) IL-13 staining. (**C-F**; **H–K**) Data representative of at least two independent experiments, with 4–5 mice/group. *p<0.05, **p<0.01, ***p<0.001 by two-tailed Welch's t-test. All data is represented as mean with scale bars representing standard deviation.

The online version of this article includes the following figure supplement(s) for figure 1:

**Figure supplement 1.** Group 2 innate lymphoid cell (ILC2) and IL-5/IL-13 gating.

**Figure supplement 2.** Expansion and activation of group 2 innate lymphoid cells (ILC2s) in recombinase activating gene (RAG2) deficiency compared to littermates.

donor ILC2s, the majority were derived from *Rag1*$^{-/-}$ donors (*Figure 2H*; *Figure 2—figure supplement 2F and G*). This was not due to differences in overall donor reconstitution, since measuring all Lin$^-$ cells revealed WT donor cells outnumbered those from *Rag1*$^{-/-}$ donors (*Figure 2—figure supplement 2E*). Of the total IL-5- (*Figure 2I*; *Figure 2—figure supplement 2H, I and K*) and IL-13-expressing (*Figure 2J*; *Figure 2—figure supplement 2H, J and L*) ILCs, the majority were also derived from *Rag1*$^{-/-}$ donors. Taken together, these data suggest that cell-intrinsic RAG activity in ILC2s may limit their capacity to expand and become activated.

## A history of RAG expression marks a subpopulation of ILC2s in the skin draining lymph node

In contrast to resting naïve T cells, ILC2s resemble activated Th2 cells at a steady state based on their transcriptomic and epigenomic profiles (*Shih et al., 2016*; *Van Dyken et al., 2016*). While both T cells and ILC2s exhibit historical RAG expression (*Yang et al., 2011*), they do not actively express the protein in their mature state (*Turka et al., 1991*). Taken together, these findings provoke the hypothesis that ILC2s are regulated by RAG early in development to imprint alterations that influence their activity as mature cells. To distinguish ILC2s as either having a history of RAG expression or not, we utilized a RAG lineage tracing system, whereby a *Rag1*$^{Cre}$ mouse was crossed to a reporter mouse expressing tandem dimer red fluorescent protein (tdRFP) in a Cre-dependent manner from the *Rosa26* locus (*Figure 3A*; *Karo et al., 2014*; *Welner et al., 2009*). This system allowed us to compare RAG-experienced (RAG$^{exp}$) and RAG-naïve (RAG$^{naïve}$) lymphoid cells, including ILC2s, simultaneously originating from the same immunocompetent host, thus removing confounders inherent in knockout and chimera experiments. Analysis of sdLN from the reporter mice revealed that nearly all CD4$^+$ T cells, CD8$^+$ T cells, and B220$^+$ B cells expressed tdRFP (positive history of Cre expression from the *Rag1* locus), consistent with the known requirement of RAG expression for their development (*Figure 3B–D and G*). We also examined NK cells, since certain subsets of NK cells are known to express RAG during their development (*Karo et al., 2014*), and we observed that roughly 60% of NK cells were tdRFP$^+$, similar to previous findings (*Figure 3E and G*; *Karo et al., 2014*). In the ILC2 population of the sdLN, around 50% were tdRFP$^+$ (*Figure 3F and G*), similar to proportions of RAG fate-mapped ILC2s previously observed in the fat (*Karo et al., 2014*) and lung (*Yang et al., 2011*; *Karo et al., 2014*). These findings demonstrate that there are heterogeneous populations of ILC2s marked by differential tdRFP$^+$ fate mapping. Importantly, this provided us with the possibility to profile these different ILC2 subsets based on *Rag1*$^{Cre}$-activated expression of tdRFP.

## Multiomic profiling enhances the detection of rare tissue-specific ILC2s

Transient RAG expression early in lymphoid development leads to well-characterized, durable effects on B and T cell development and function mainly through successful genomic rearrangement of antigen receptors. Yet, our data indicate that RAG expression also imprints phenotypic changes on ILC2s, which can develop independently of functional antigen receptors, provoking the hypothesis that RAG expression may affect broader epigenomic and transcriptional programs. Furthermore, our data indicate that the impact of RAG on ILC2 function has implications for AD-like skin inflammation, suggesting a persistent effect that modulates phenotypes of type 2 inflammation.

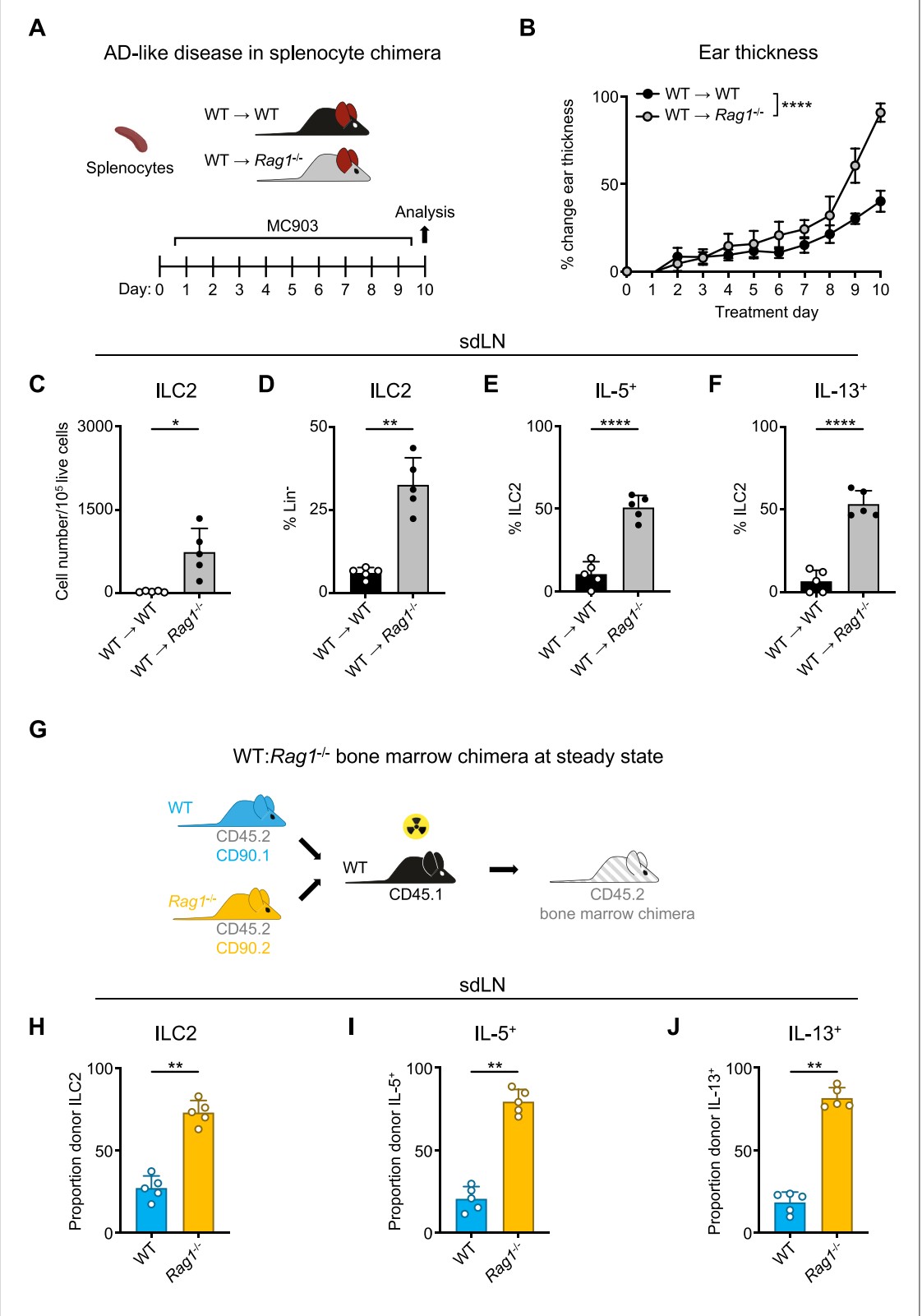

**Figure 2.** Homeostatic expansion and activation of recombinase activating gene (RAG)-deficient group 2 innate lymphoid cells (ILC2s) is cell intrinsic. (**A**) Experimental schematic of atopic dermatitis (AD)-like disease in splenocyte chimera experiment. Wild-type (WT) B6 or *Rag1⁻/⁻* mice received WT splenocytes and developed AD-like inflammation after subsequent topical treatment with 2 nmol MC903 in 10 μL ethanol vehicle to each ear daily for 10 days. (**B**) Ear thickness measured daily in AD-like inflammation. Data representative of two independent experiments, 4–5 mice per group.

*Figure 2 continued on next page*

*Figure 2 continued*

****p<0.0001 by two-way ANOVA with Sidak's multiple comparisons test, day 10. (**C**) Total number ILC2s normalized to $10^5$ live cells and (**D**) proportion of CD90$^+$, Lin$^-$ cells (Lin$^-$ defined as CD3$^-$, CD5$^-$, CD11b$^-$, CD11c$^-$, CD19$^-$, NK1.1$^-$, and FcεR1$^-$) determined to be ILC2s (IL-33R$^+$). Percent ILC2 from skin-draining lymph node (sdLN) in splenocyte chimera mice with AD-like disease after Phorbol 12-myristate 13-acetate (PMA)/ionomycin stimulation positive for (**E**) IL-5 or (**F**) IL-13 staining. (**G**) Schematic of bone marrow chimera experiment. Equal quantities of bone marrow cells from *Rag1$^{-/-}$* (CD45.2, CD90.2 - orange) and WT (CD45.2, CD90.1 - blue) C57BL/6J donor mice were used to reconstitute the immune systems of irradiated recipient WT (CD45.1 - black) C57BL/6J mice. (**H**) Proportion of donor (CD45.2$^+$) ILC2 defined as in (**C and D**) in sdLN by donor source (CD90.1$^+$ - WT, CD90.2$^-$ - *Rag1$^{-/-}$*). Proportion of Lin$^-$ ILCs by donor source positive for (**I**) IL-5 and (**J**) IL-13 following PMA/ionomycin stimulation and cytokine staining. (**C–F**) Data representative of at least two independent experiments, 4–5 mice per group. **p<0.01, ****p<0.0001 by two-tailed Welch's t test. (**H–J**) Data representative of at least two independent experiments with 4–5 mice per group. **p<0.01 by two-tailed ratio means paired t test. All data represented as mean with scale bars representing standard deviation.

The online version of this article includes the following figure supplement(s) for figure 2:

**Figure supplement 1.** Confirmation of splenocyte reconstitution in splenocyte chimera mice.

**Figure supplement 2.** Donor cell reconstitution and gating in sdLN of wild-type (WT):*Rag1$^{-/-}$* bone marrow chimera mice.

To test these hypotheses, we performed combined single-nuclei RNA sequencing (snRNA-seq) and ATAC sequencing (snATAC-seq) of sdLN cells (*Figure 4*) from RAG fate-mapped mice at steady state and in the setting of AD-like disease (*Figure 4—figure supplement 1A and B*). Because fluorescence-activated cell sorting (FACS) can cause physical stress, cell loss, and contamination, which can introduce unwanted perturbations in target cells, instead we utilized gentle initial negative selection by magnetic activated cell sorting (MACS) to remove most B and T cells and monocytes prior to sequencing (*Figure 4—figure supplement 1A*). This allowed us to enrich for innate immune cell populations prior to sequencing. Further, we used the gene encoding tdRFP as a barcode to differentiate between RAG$^{exp}$ (RAG fate map-positive) and RAG$^{naïve}$ (RAG fate map-negative) ILC2s at the single-cell level (*Figure 5A*). The multiomic data was analyzed using recently developed pipelines in Cell Ranger, Seurat (*Stuart et al., 2019*; *Hao et al., 2021*; *Butler et al., 2018*), and Signac (*Stuart et al., 2021*), and sequenced cells were further filtered computationally to enrich for ILCs, as in previous studies (see methods) (*Ghaedi et al., 2020*).

In addition to gene expression (GEX) information derived from snRNA-seq (*Figure 4A*), we calculated a 'gene activity' (GA) score based on chromatin accessibility at gene loci (*Figure 4B*) from the corresponding snATAC-seq dataset (*Stuart et al., 2021*). Clustering the cells with each data subset alone and in combination using weighted nearest neighbor (WNN) analysis, we identified six clusters (*Figure 4C*) that demonstrated consistent differences in cellular markers based on both metrics of GEX and GA (*Figure 4D and E*; *Supplementary file 1*, Table S1). Additionally, top markers for each cluster clearly differentiated each cell type (*Figure 4F*). Despite successful ILC2 enrichment via MACS depletion for lineage markers and computational filtering (see methods), our data set included non-ILC2 populations determined by gene expression to be T cells, dendritic cells, B cells, and NK cells (*Figure 4C and F*; *Supplementary file 1*, Table S1), which allowed for broader multidimensional comparisons while studying this ILC2-enriched data set.

To further complement the GEX and GA assays, we utilized another method of detecting cell-specific marker genes, whereby chromatin regions that are differentially accessible (DA), or open, in each cluster could be linked by their physical proximity to specific genes (*Figure 4G*; *Supplementary file 1*, Table S2; see methods). Comparing the top 100 GEX, GA, and DA markers in the ILC2 population, we identified a multiomic ILC2 signature of 235 unique genes (*Figure 4H*; *Supplementary file 1*, Table S3, S4). While there was some overlap between each respective set, the multiomic approach enabled more extensive identification of an ILC2 gene program than through either snRNA-seq or snATAC-seq alone. The analysis revealed a variety of canonical ILC2-associated genes specific to the ILC2 cluster (*Figure 4I*, *Figure 4—figure supplement 2*, *Figure 4—figure supplement 3*) including the ILC2-activating cell surface receptors *Icos* (*Maazi et al., 2015*; *Paclik et al., 2015*), *Il2ra* (*Roediger et al., 2013*; *Roediger et al., 2015*), *Il18r1* (*Ghaedi et al., 2020*; *Zeis et al., 2020*; *Ricardo-Gonzalez et al., 2018*), *Nmur* (*Tsou et al., 2022*; *Jarick et al., 2022*; *Cardoso et al., 2017*; *Klose et al., 2017*; *Wallrapp et al., 2017*; *Figure 4—figure supplement 2A*), and *Il1rl1* (encoding the receptor for IL-33) (*Imai et al., 2013*; *Salimi et al., 2013*; *Mjösberg et al., 2011*; *Hung et al., 2013*), as well as transcription factors such as *Gata3* (*Moro et al., 2010*; *Mjösberg et al., 2012*; *Hoyler et al., 2012*; *Klein Wolterink et al., 2013*; *Pokrovskii et al., 2019*), *Bcl11b* (*Califano et al., 2015*; *Walker et al., 2015*;

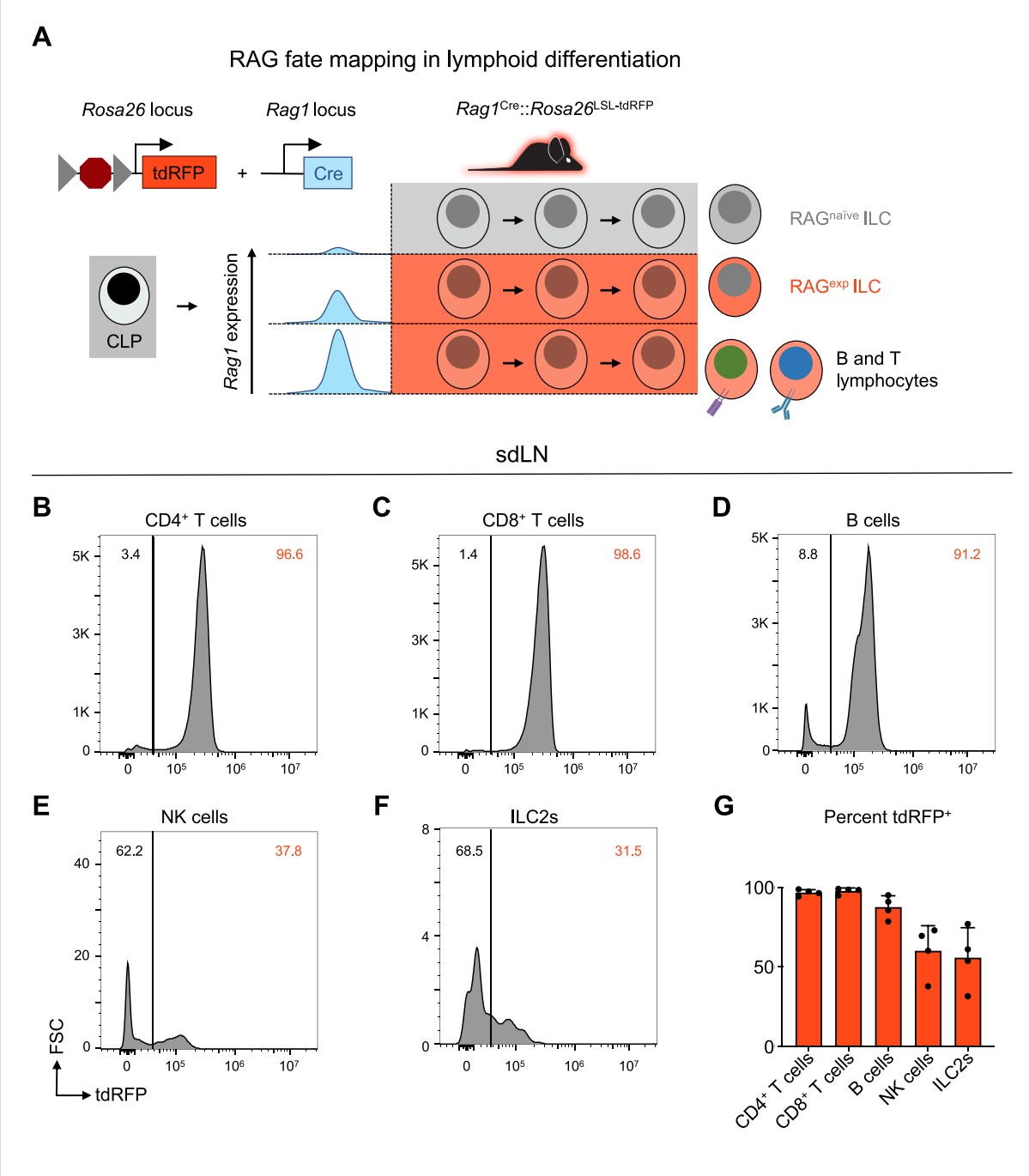

**Figure 3.** A history of recombinase activating gene (RAG) expression marks a population of group 2 innate lymphoid cells (ILC2s) in the skin-draining lymph node (sdLN). (**A**) Schematic of RAG fate mapping in the lymphoid cell compartment using reporter mice expressing Cre-inducible tandem dimer red fluorescent protein (tdRFP) from the *Rosa26* locus crossed to mice expressing Cre recombinase from the *Rag1* locus. (**B–F**) Histograms of tdRFP signal in CD45+ sdLN cells by cell type for (**B**) CD4+ T cells (B220-, CD3+, CD4+), (**C**) CD8+ T cells (B220-, CD3+, CD8+), (**D**) B cells (MHCII+, B220+), (**E**) NK cells (B220-, CD3-, CD4-, CD8-, CD49b+, NK1.1+), (**F**) ILC2s (B220-, CD3-, CD4-, CD8-, CD49b-, NK1.1-, CD11b-, CD11c-, SiglecF-, CD90+, KLRG1+ or ICOS+ or IL-33R+), (**G**) quantification of tdRFP+ proportion of each cell type. Data representative of at least two independent experiments, 3-4 mice per group. Data in (**G**) represented as mean with scale bars representing standard deviation. Statistical analyses were not performed.

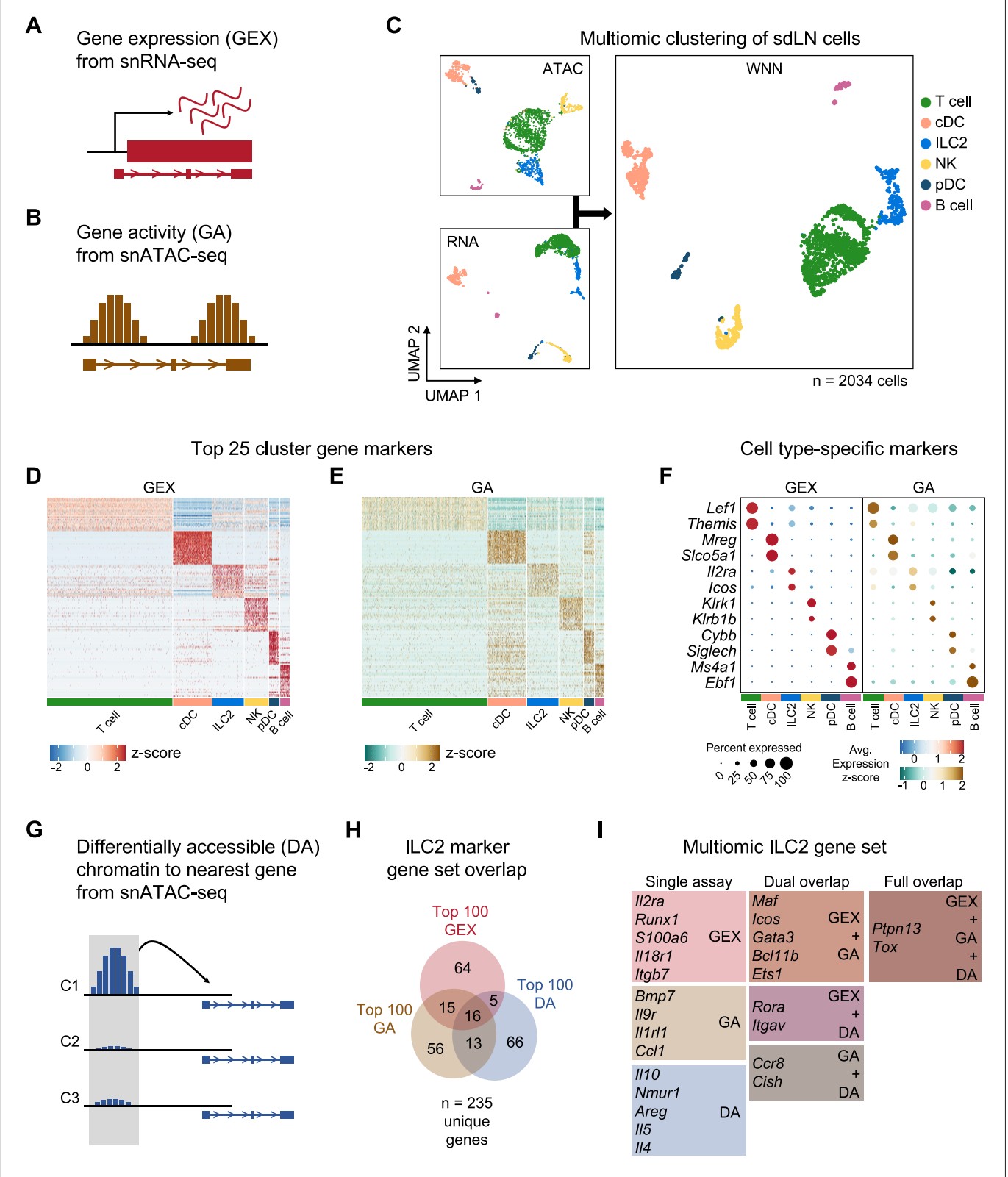

**Figure 4.** Multiomic analysis of group 2 innate lymphoid cells (ILC2s) through single nuclei sequencing of the skin-draining lymph node (sdLN). (**A**) Schematic of the gene expression (GEX) assay derived from single-nuclei RNA sequencing (snRNA-seq) data. (**B**) Schematic of the gene activity (GA) assay, representing estimated transcription scores derived from single-nuclei ATAC sequencing (snATAC-seq) data using Signac. (**C**) UMAP visualizations of independent analyses of RNA-seq and ATAC-seq data for 2034 sdLN cells after dimensional reduction and clustering combined using weighted

*Figure 4 continued on next page*

*Figure 4 continued*

nearest neighbor (WNN) analysis in Seurat. Cluster identities are color coded consistently throughout the following panels. Heatmaps of (**D**) top 25 GEX marker genes and (**E**) top 25 GA marker genes identified for each cluster. See *Supplementary file 1*, Table S1 for full lists of genes. (**F**) Dotplots comparing selected marker genes for each cluster between the GEX and GA assays, with emphasis on known cell type-specific markers. (**G**) Schematic of differentially accessible (DA) chromatin assay, which finds the nearest gene to any peak calculated to be differentially open in a particular cell cluster. See *Supplementary file 1*, Table S2 for full lists of the top 25 DA cluster markers. (**H**) Overlap of top 100 markers for the ILC2 cluster from the GEX, GA, and DA assays. See *Supplementary file 1*, Table S3 for the top 100 DA peaks and distances to the nearest genes and *Supplementary file 1*, Table S4 for the full list of the top 100 ILC2 markers. (**I**) Selected genes from the ILC2 gene set for each assay individually and for overlaps.

The online version of this article includes the following figure supplement(s) for figure 4:

**Figure supplement 1.** Skin-draining lymph node (sdLN) multiome experiment.

**Figure supplement 2.** Group 2 innate lymphoid cells (ILC2) marker genes were identified in the differentially accessible open chromatin assay.

**Figure supplement 3.** Dotplots of selected group 2 innate lymphoid cells (ILC2) marker genes.

*Yu et al., 2015*; *Hosokawa et al., 2020*), Maf (*Moro et al., 2010*; *Björklund et al., 2016*; *Trabanelli et al., 2022*), Ets1 (*Zook et al., 2016*), and Rora (*Ghaedi et al., 2020*; *Hoyler et al., 2012*; *Wong et al., 2012*; *Halim et al., 2012*; *Ferreira et al., 2021*), all previously shown to be important for ILC2 development and/or function.

Expression of some secreted proteins can be difficult to capture in droplet-based snRNA-seq experiments due to low transcript levels and relatively shallow sequencing depth. With the addition of the complementary GA and DA assays from snATAC-seq, our analysis identified *Il5* (*Figure 4I*, *Figure 4—figure supplement 2B*), a canonical ILC2 cytokine, in the DA assay, while in the GA assay, we found *Bmp7* (*Miyajima et al., 2020*; *Chen et al., 2021*), which has been shown to be secreted by ILC2s to influence browning of adipose tissue. Additionally, we identified the secreted chemokine *Ccl1* as an ILC2 marker gene (*Xu et al., 2019*; *Vivier et al., 2016*; *Schneider et al., 2019*; *Bielecki et al., 2021)*, which along with its cognate receptor *Ccr8* (also an ILC2 marker in our analysis) (*Ricardo-Gonzalez et al., 2018*; *Vivier et al., 2016*; *Sun et al., 2022*), participates in a feed-forward circuit to drive ILC2 recruitment and expansion (*Knipfer et al., 2019*). Thus, our findings demonstrate how genetic barcoding, combining transcriptomic and epigenomic analyses, and cross-validation across many published studies can yield new insights while providing internal control measures to elevate the rigor, robustness, and confidence of identifying gene signatures of rare populations such as ILC2s at the single-cell level.

## ILC2s with a history of RAG expression are epigenomically suppressed

As noted above, barcoding the ILC2s afforded the opportunity to transcriptionally and epigenomically profile ILC2s under identical developmental conditions by dividing the ILC2 cluster into RAG^exp and RAG^naïve populations (*Figure 5A*). We hypothesized that RAG^exp ILC2s would have a distinct transcriptional profile compared to ILC2s without any history of RAG expression. To test this, we calculated differentially expressed genes (DEGs) for the ILC2 cluster by RAG fate-map status. Genes with higher expression in RAG^exp cells relative to RAG^naïve cells had positive fold change values, and vice versa, with genes relatively increased in RAG^naïve cells having negative values (*Figure 5—figure supplement 1A*, *Supplementary file 1*, Table S5). Using gene set enrichment analysis (GSEA) *Subramanian et al., 2005*; *Mootha et al., 2003* on the ranked list of DEGs, we found that gene sets generally representing lymphocyte activation and differentiation were suppressed in RAG^exp ILC2s compared to RAG^naïve ILC2s (*Figure 5—figure supplement 1B and C*, *Supplementary file 1*, Table S6), consistent with our previous observations that ILC2s are expanded and more activated in RAG-deficient mice relative to WT mice.

We next employed newly described methodologies (*Stuart et al., 2021*; *Ma et al., 2020*) that quantify associations between open chromatin peaks and the expression of nearby genes to describe the functional regulomes of both RAG^exp and RAG^naïve ILC2s (*Figure 5B*). In this analysis, each ATAC peak can be linked to multiple genes, and each gene to multiple peaks, generating a list of 'gene-to-peak links' or GPLs (see methods). For each gene, we interpreted the number of corresponding GPLs as a quantitative representation of the regulome activity for that gene. Considering RAG^exp and RAG^naïve cells as two separate populations, we generated two lists of GPLs (*Supplementary file 1*, Table S7) defining functional regulomes for each population. We focused our analysis on the functional regulomes of ILC2s by filtering the GPL lists for the 235 unique ILC2 genes identified in our

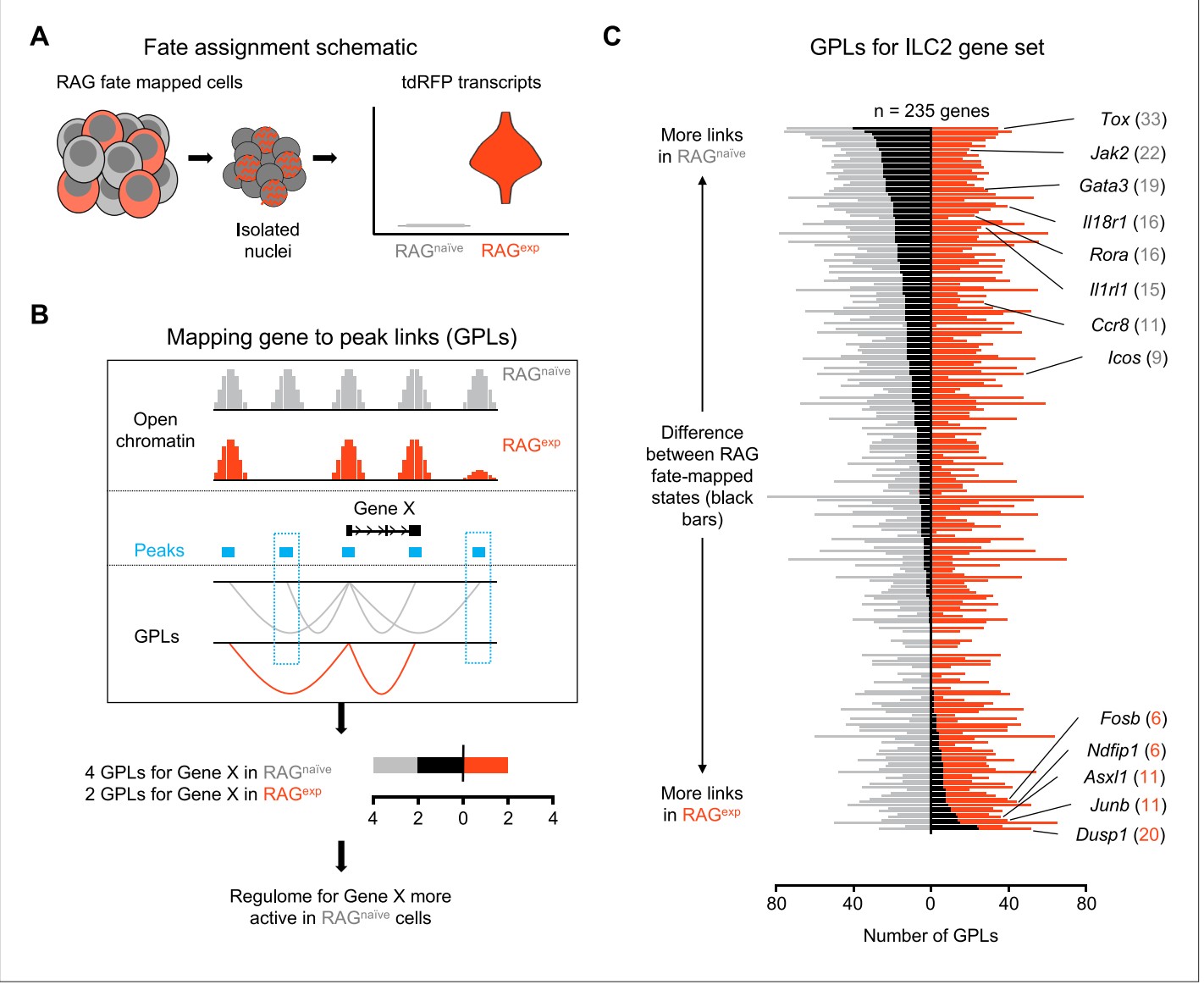

**Figure 5.** A history of recombinase activating gene (RAG) expression imprints transcriptomic and epigenomic modulation of group 2 innate lymphoid cells (ILC2) gene programs. (**A**) Schematic of transcriptional RAG fate mapping. Sequenced cells from the RAG fate map mouse (see *Figure 3A*) transcribe tdRFP only after Cre is expressed from the *Rag1* locus. Cells were assigned as either having a history of RAG expression (RAG^exp - tomato red) or not (RAG^naïve - dark gray) based on the detection of tdRFP transcript in the RNA-seq data (*Supplementary file 1*, tdRFP sequence). (**B**) Schematic of mapping gene-to-peak links (GPLs). The LinkPeaks function of Signac (see methods) calculates significant correlations between open chromatin at defined peaks (teal bars) and nearby gene expression. These links represent inferred epigenomic-transcriptomic regulation, or 'regulomes' based on the correlated single-nuclei RNA (snRNA)- and single-nuclei ATAC (snATAC)-sequencing data. After calculating GPLs separately for each population (gray for RAG^naïve and tomato red for RAG^exp), GPLs found in only one group, but not the other, can be identified (teal boxes). The difference in GPLs based on RAG experience for any given gene (e.g. Gene X) can be visualized on a bar graph, with the number of GPLs for RAG^naïve (gray - left) and RAG^exp (red - right) plotted and the difference overlaid as a black bar. (**C**) GPLs calculated as in (**B**) for the multiomic ILC2 gene set identified in *Figure 4H* and (*Supplementary file 1*, Table S4). All identified GPLs are listed in *Supplementary file 1*, Table S7, while ILC2 GPLs are listed in *Supplementary file 1*, Table S8. Genes are sorted from more links identified in the RAG^naïve population at the top to more links identified in the RAG^exp population at the bottom. Select genes are labeled. Full ranked list by the difference in GPLs is available in *Supplementary file 1*, Table S9.

The online version of this article includes the following figure supplement(s) for figure 5:

**Figure supplement 1.** Gene set enrichment analysis of differentially expressed genes in group 2 innate lymphoid cells (ILC2s).

**Figure supplement 2.** Mapping gene-to-peak links in select group 2 innate lymphoid cell (ILC2) genes.

**Figure supplement 3.** Multiomic transcription factor analysis of group 2 innate lymphoid cells (ILC2s).

multiomic gene set (*Figure 4H*; *Supplementary file 1*, Table S8), then calculated the difference in GPLs between RAG^exp and RAG^naïve cells for each gene and sorted them; genes displaying greater numbers of GPLs in the RAG^naïve population are at the top, and genes with more GPLs in the RAG^exp population are at the bottom (*Figure 5C*; *Supplementary file 1*, Table S9).

We found that the ILC2 marker genes segregating toward the top of this list, corresponding to enhanced epigenomic activity in the RAG^naïve cells, tended to be genes previously identified to play positive roles in the development, expansion, and activation of lymphoid cells. These included transcriptional regulators such as *Tox* (*Constantinides et al., 2014*; *Seehus et al., 2015*; *Aliahmad et al., 2010*; *Aliahmad and Kaye, 2008*), *Rora* (*Ghaedi et al., 2020*; *Hoyler et al., 2012*; *Wong et al., 2012*; *Halim et al., 2012*; *Ferreira et al., 2021*), *Maf* (*Moro et al., 2010*; *Björklund et al., 2016*; *Trabanelli et al., 2022*), and *Gata3* (*Moro et al., 2010*; *Mjösberg et al., 2012*; *Hoyler et al., 2012*; *Klein Wolterink et al., 2013*; *Figure 5—figure supplement 2A*)**,** which are involved in early differentiation of both ILCs and lymphocytes. Indeed, epigenetic activation of the *Gata3* locus is recognized to play a critical role in the development of both ILC2s (*Kasal et al., 2021*) and Th2 cells (*Wei et al., 2010*; *Onodera et al., 2010*). Additionally, surface receptors known to drive ILC2 activation upon stimulation including *Il18r1* (*Ghaedi et al., 2020*; *Zeis et al., 2020*; *Ricardo-Gonzalez et al., 2018*), *Il1rl1* (*Neill et al., 2010*; *Ghaedi et al., 2020*; *Zeis et al., 2020*; *Barlow et al., 2013*; *Molofsky et al., 2015*), and *Icos* (*Maazi et al., 2015*; *Paclik et al., 2015*) had increased functional regulome activity in RAG^naïve ILC2s. In contrast, genes with more GPLs in the RAG^exp ILC2s tended to be associated with suppressive functions. For example, *Ndfip1* (*Figure 5—figure supplement 2B*) encodes a regulatory protein that enhances the activity of the ubiquitin ligase ITCH to negatively regulate inflammation (*Oliver et al., 2006*; *Altin et al., 2014*) and has been associated with asthma risk in GWAS studies (*Ferreira et al., 2011*). *Dusp1* partially mediates glucocorticoid effects through its ability to negatively regulate inflammation (*Chi et al., 2006*; *Clark et al., 2008*), is associated with eczema by GWAS (*Grosche et al., 2021*), and has recently been shown to mark an anti-inflammatory set of ILCs (*Bielecki et al., 2021*). Last, *Asxl1* encodes a tumor suppressor that inhibits clonal hematopoiesis through its epigenomic regulatory effects in both mice and humans (*Xie et al., 2014*; *Genovese et al., 2014*; *Siddhartha et al., 2014*; *Nagase et al., 2018*). Collectively, our GPL analysis stratifies the ILC2 gene signature based on RAG experience, where genes associated with ILC2 expansion and activation are poised in RAG^naïve cells, while genes associated with suppressive effects are poised in RAG^exp cells.

We expanded our multiomic analysis to infer information about transcription factor (TF) activity from open chromatin regions in our snATAC data. We used the chromVAR package (*Schep et al., 2017*), which finds known TF binding motifs in open chromatin regions in each cell, to identify TF motifs enriched in each cell cluster (*Figure 5—figure supplement 3A and B*, *Supplementary file 1*, Table S10). The enriched motifs were consistent with the known functional roles of associated TFs in each cell type. For example, in the NK cell cluster, we found enriched motifs recognized by the TFs EOMES and T-bet (encoded by *Eomes* and *Tbx21*, respectively), which are critical for the development of NK cells (*Bando and Colonna, 2016*). A limitation of this analysis is that while TF motif accessibility can be inferred from open chromatin in snATAC data, which TFs are bound to the identified accessible sites is not known. We reasoned that complementary gene expression information from our multiomic data could mitigate this limitation in part by comparing the accessibility of TF binding motifs to expression levels of corresponding TFs (*Figure 5—figure supplement 3C*). Indeed, motifs for both RORα and RORγ (encoded by *Rora* and *Rorg*, respectively), which share a common DNA binding 5'-AGGTCA-3' half site, have similar calculated accessibilities in both the ILC2 and NK cell clusters. Yet only *Rora* is expressed at appreciable levels, and only in ILC2s, consistent with its critical role in ILC2 development (*Wong et al., 2012*; *Halim et al., 2012*). In contrast, ILC2 development is not dependent on *Rorg* expression, and neither RORα nor RORγ plays a major role in NK cells. Taken together, this analysis confirms the known role of *Rora* in ILC2s and highlights how matched multiomic chromatin accessibility and gene expression data can clarify ambiguities inherent in TF enrichment analyses.

The broad effects of RAG expression on ILC2 transcriptional regulomes we observed (*Figure 5C*) led us to hypothesize that distinct cohorts of TFs may contribute to the differences observed between RAG^naïve and RAG^exp ILC2s. To test this hypothesis, we analyzed the open chromatin regions in GPLs unique to each RAG fatemapped ILC2 population using the FindMotifs function in Signac (*Stuart et al., 2021*), which returns a ranked list of enriched motifs corrected for the background presence of

each motif in all cells. In both RAG^naïve and RAG^exp ILC2s, we identified enriched TF motifs (*Figure 5—figure supplement 3D*, *Supplementary file 1*, Table S11) that are GC-rich regions recognized by a large family of C2H2 zinc finger TFs, particularly the Krüppel-like factors (KLFs), which are well-established as key regulators of lymphocyte development (*Hart et al., 2012*; *Cao et al., 2010*). Given the strong sequence similarities of the identified TF motifs, we turned to the matched gene expression data to clarify which TFs may be available to engage the accessible binding sites. Of the eleven unique TFs identified, only six were detected in the gene expression assay (*Figure 5—figure supplement 3E*). We observed much higher expression of *Klf2*, *Klf6*, and *Klf12* in Rag^exp ILC2s compared to RAG^naïve ILC2s (*Figure 5—figure supplement 3E*). Notably, all three of these TFs have been associated with reduced cellular proliferation and/or activation (*Weinreich et al., 2009*; *Godin-Heymann et al., 2016*; *Narla et al., 2001*). *Klf2* expression plays a key role in T cell quiescence (*Kuo et al., 1997*; *Buckley et al., 2001*), and both *Klf2* and *Klf6* were recently identified as markers of 'quiescent-like' skin resident ILCs (*Bielecki et al., 2021*). In contrast, although detected in a smaller fraction of cells in our data, we found *Klf7* expression was higher in RAG^naïve ILC2s compared to Rag^exp ILC2s (*Figure 5—figure supplement 3E*). Increased expression of *Klf7* has been shown to enhance the survival of early thymocytes and is a predictor of poor outcomes in acute lymphoblastic leukemia (*Schuettpelz et al., 2012*). Collectively, these findings link the relatively activated or suppressed epigenomic and transcriptomic states of RAG^naïve and Rag^exp ILC2s, respectively, to distinct cohorts of homeostatic TFs.

## A history of RAG expression modulates ILC2 epigenomes at a steady state and in AD-like inflammation

Although our GPL and TF analyses revealed a suppressive effect of RAG expression on ILC2 gene programs, we did not account for the additional variable of disease state in the initial analysis. To test whether RAG expression promotes a suppressive epigenomic program in ILC2s that is durable in the setting of inflammation, we first recalculated GPLs after splitting our dataset by both history of RAG expression (naïve vs. experienced) and disease (steady-state vs. AD-like disease) to yield four lists of GPLs (*Figure 6A*; *Supplementary file 1*, Table S12). When we examined the intersection, or overlap, of peaks from ILC2 GPLs (*Supplementary file 1*, Table S13), several notable patterns emerged (*Figure 6B*). First, the largest set of peaks was shared by all RAG^naïve cells (gray bar), regardless of disease state, with the next largest peak sets belonging to either steady state or AD-like disease in the RAG^naïve cells. Second, there was a large set of peaks shared by all RAG^exp cells (red bar). Third, the intersections corresponding to disease states (steady state – yellow, AD-like disease – dark red), had relatively few unique peaks. These findings suggest that early exposure to RAG expression plays a larger role in modulating the epigenomic signature of the ILC2 gene program than exposure to disease. To confirm that the patterns we observed represent a specific effect of RAG expression on the ILC2 gene program, we performed the same analysis on GPL peaks for all genes in the dataset (*Figure 6—figure supplement 1*). In contrast to the ILC2 gene set, the majority of GPL peaks for all genes was shared among all cell populations, consistent with epigenomic regulation of most genes being minimally affected by either RAG expression or AD-like disease. Last, in the ILC2 gene set analysis, we noted a set of poised peaks shared by all RAG^naïve cells and RAG^exp cells in the setting of AD-like disease, but not with RAG^exp cells at steady state (blue bar, *Figure 6B*). We reasoned that this condition might capture some genomic loci that are suppressed by a history of RAG expression at steady state but are induced during inflammation.

Thus, we next quantified and sorted these GPLs to generate a list of genes with the most peaks 'induced' during AD-like disease (*Figure 6C*, *Supplementary file 1*, Table S14). Among the identified genes, we selected *Rora* (*Figure 6D*) and *Ccr6* (*Figure 6E*) to examine more closely for evidence of epigenomic activation in AD-like disease, given the role of these genes in ILC2 expansion (*Wong et al., 2012*; *Halim et al., 2012*) and homing to sites of inflammation (*Ricardo-Gonzalez et al., 2018*; *Kobayashi et al., 2019*), respectively. For both genes, we observed more widespread open chromatin over the genomic region in the RAG^naïve cells compared to the RAG^exp cells, but this difference was partially abolished by increased open chromatin in AD-like disease in the RAG^exp cells. Taken together, our analysis reveals that a history of RAG expression selectively modulates the activity of ILC2 gene programs across both steady state and during AD-like inflammation, while some programs are more evident at steady state given the uniquely poised nature of ILC2s.

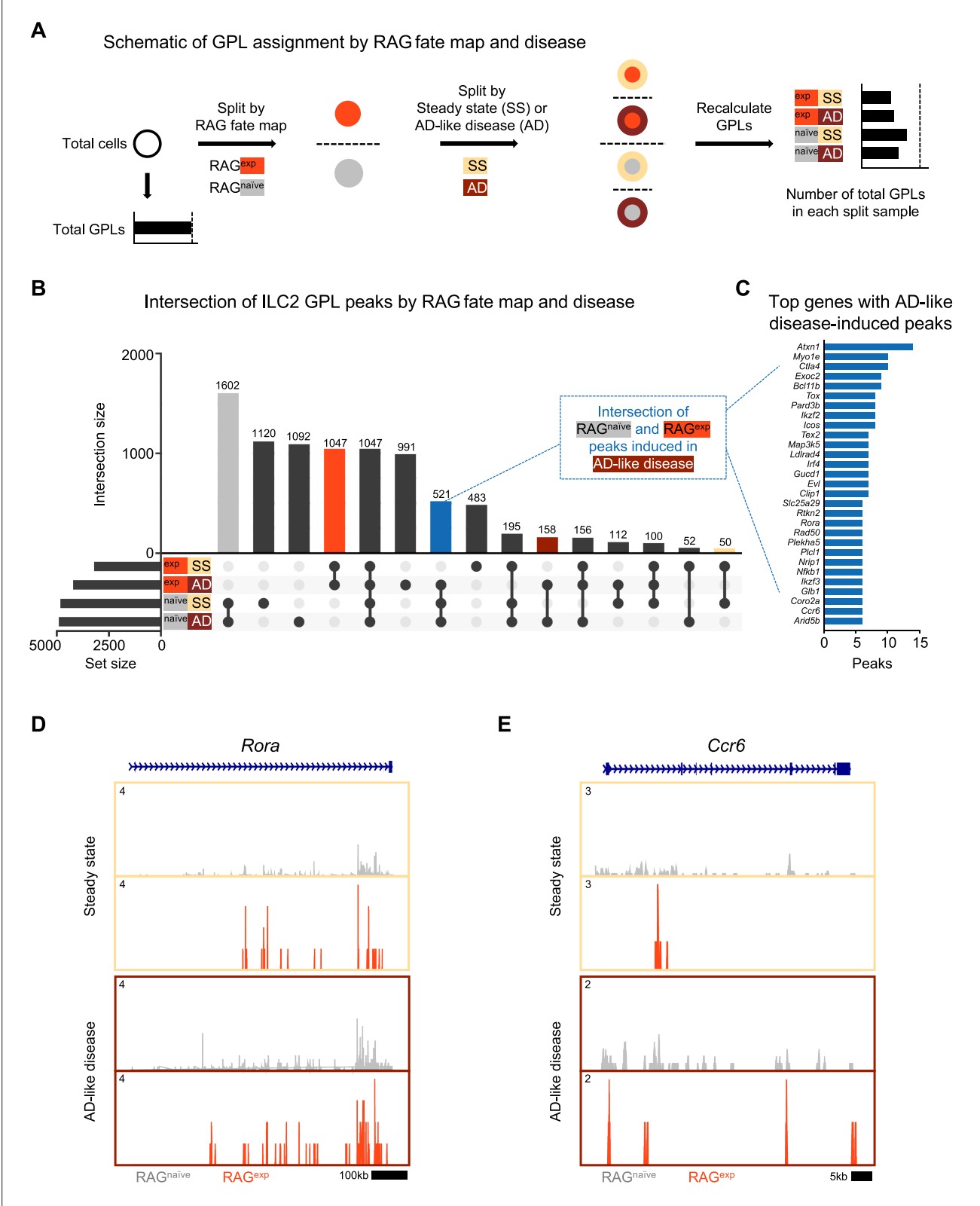

**Figure 6.** A history of recombinase activating gene (RAG) expression broadly influences group 2 innate lymphoid cell (ILC2) genes at steady state and in atopic dermatitis (AD)-like inflammation. (**A**) Schematic of the process to determine the contribution of RAG fate map and disease states to gene-to-peak links (GPLs) for subsequent intersection analyses. GPLs were first calculated for all indicated cells, regardless of the disease state or fate map (see methods). Cells were then split, first by RAG fate map (RAGexp and RAGnaïve), and again by disease state (SS - steady state, AD - atopic dermatitis-like

*Figure 6 continued on next page*

*Figure 6 continued*

inflammation). GPLs were recalculated for each split sample and matched back to the original set of total GPLs. (**B**) UpSet plot visualizing intersections of peaks identified from ILC2 GPLs for split samples. Each row represents one of the four sets, and each column corresponds to an intersection of one or more sets (see methods). See ***Supplementary file 1***, Table S12 for the full list of GPLs for all genes. ***Supplementary file 1***, Table S13 lists the total and ILC2 peaks used for intersection analyses in each of the four sets. Columns identifying key intersections are color coded by the corresponding RAG fate map or treatment groups. The blue column indicates the intersection of peaks from RAG$^{naïve}$ cells and peaks induced by AD-like disease in RAG$^{exp}$ cells. (**C**) Top genes with the most AD-like disease-induced peaks. Peaks from the intersection between RAG$^{naïve}$ cells and inflamed RAG$^{exp}$ cells were identified in corresponding GPLs, and genes were ranked by the number of linked peaks identified. See ***Supplementary file 1***, Table S14 for the full list of ranked genes and associated GPLs. Open chromatin in the ILC2 cell cluster split by disease (beige box – steady state; maroon box – AD-like disease) and by RAG fate map (RAG$^{naïve}$ - gray, RAG$^{exp}$ - red) for the genomic loci of (**D**) *Rora* and (**E**) Ccr6.

The online version of this article includes the following figure supplement(s) for figure 6:

**Figure supplement 1.** Gene-to-peak link analysis by recombinase activating gene (RAG) fate map and disease for all detected genes.

## RAG suppresses the Th2 locus

Our functional data demonstrate a role for RAG expression in regulating ILC2 development and activation, including limiting proportions of IL5$^+$ and IL-13$^+$ ILC2s at steady state and in AD-like disease. Prior work identified epigenomic priming in ILC2s early in development at the Th2 locus (comprised of the *Il4*, *Il13*, *Rad50*, and *Il5* gene loci) to enable rapid transcriptional responses during inflammation (***Shih et al., 2016***). Thus, we hypothesized that RAG promotes the functional observations in ILC2s by suppressing the establishment of an active regulome at the Th2 locus. To test this hypothesis, we analyzed the Th2 locus in our multiomic data in greater detail. Using a similar strategy to our analysis of ILC2 marker genes, we calculated the number of GPLs in the RAG$^{exp}$ and RAG$^{naïve}$ cells, respectively, for the genes in the Th2 locus. (***Figure 7A***, ***Supplementary file 1***, Table S7). We found many GPLs associated with the four Th2 locus genes, including significant crosstalk *between* these genes, similar to previous observations (***Figure 7A***; ***Lee et al., 2003***; ***Loots et al., 2000***; ***Fields et al., 2004***). Importantly, we identified fewer GPLs in the RAG$^{exp}$ cells, particularly for the *Il5* and *Il13* loci (***Figure 7A***). As in our analysis of the ILC2 marker GPLs (***Figure 5***), we quantified the differences based on RAG fate mapping and found that all genes in this locus had increased GPLs in RAG$^{naïve}$ cells relative to RAG$^{exp}$ cells (***Figure 7B***; ***Supplementary file 1***, Table S15). We also applied the same analysis strategy that identified TFs potentially mediating observed differences between RAG$^{exp}$ and RAG$^{naïve}$ ILC2s (***Figure 5—figure supplement 3D and E***) specifically to the four Th2 locus genes. Given the limited size of genomic regions (and thus open chromatin peaks) analyzed in the Th2 locus compared to all ILC2 genes, we found overall fewer enriched motifs. Strikingly, significant enrichment of TF motifs was only present in unique peaks from RAG$^{naïve}$ ILC2s, while no TF motifs met the cutoff in RAG$^{exp}$ cells (***Figure 7—figure supplement 1A***, ***Supplementary file 1***, Table S16). These motifs primarily contained the canonical 5'-(A/T)GATA(A/G)–3' binding site recognized by the GATA family of zinc finger TFs (***Lentjes et al., 2016***). When we compared enriched motifs in open chromatin to gene expression of the corresponding TFs, only *Gata3* was expressed at appreciable levels (***Figure 7—figure supplement 1B***). Critically, *Gata3* expression was higher in RAG$^{naïve}$ compared to RAG$^{exp}$ cells, consistent with our previous analyses of the ILC2 gene regulomes (***Figure 5C***). Collectively, our data confirm the established role of GATA3 in mediating activation of the Th2 locus (***Mjösberg et al., 2012***) and are consistent with a role for RAG expression in suppressing the type 2 regulome at the Th2 locus.

We next considered the additional effects of AD-like inflammation on the Th2 epigenomic regulome using the same approach we used to analyze the ILC2 gene set in ***Figure 6***. Again, we found the largest set of peaks was shared by the RAG$^{naïve}$ cells, regardless of disease state, with the next largest peak sets belonging to either steady state or AD-like disease in the RAG$^{naïve}$ cells (***Figure 7C***). Furthermore, there was a large proportion of peaks shared by both RAG$^{exp}$ cells, consistent with a major contribution of a history of RAG expression to epigenomic modulation of the Th2 regulome. To quantify the potential effect of AD-like inflammation on reversing RAG-mediated suppression of Th2 locus genes, we mapped the 14 peaks shared by RAG$^{naïve}$ cells and RAG$^{exp}$ cells in the setting of AD-like inflammation (i.e. only suppressed in RAG$^{naïve}$ cells at steady state) back to the Th2 genes via their respective GPLs (***Figure 7C*** - blue bar, ***Supplementary file 1***, Table S17). Interestingly, *Il13*, which was not identified as a top ILC2 marker in our earlier analyses, had the highest number of linked peaks associated with potential induction in AD-like disease (***Figure 7D***). When we examined the *Il13* locus in the ILC2 cluster more closely, we found more widespread open chromatin in the RAG$^{naïve}$

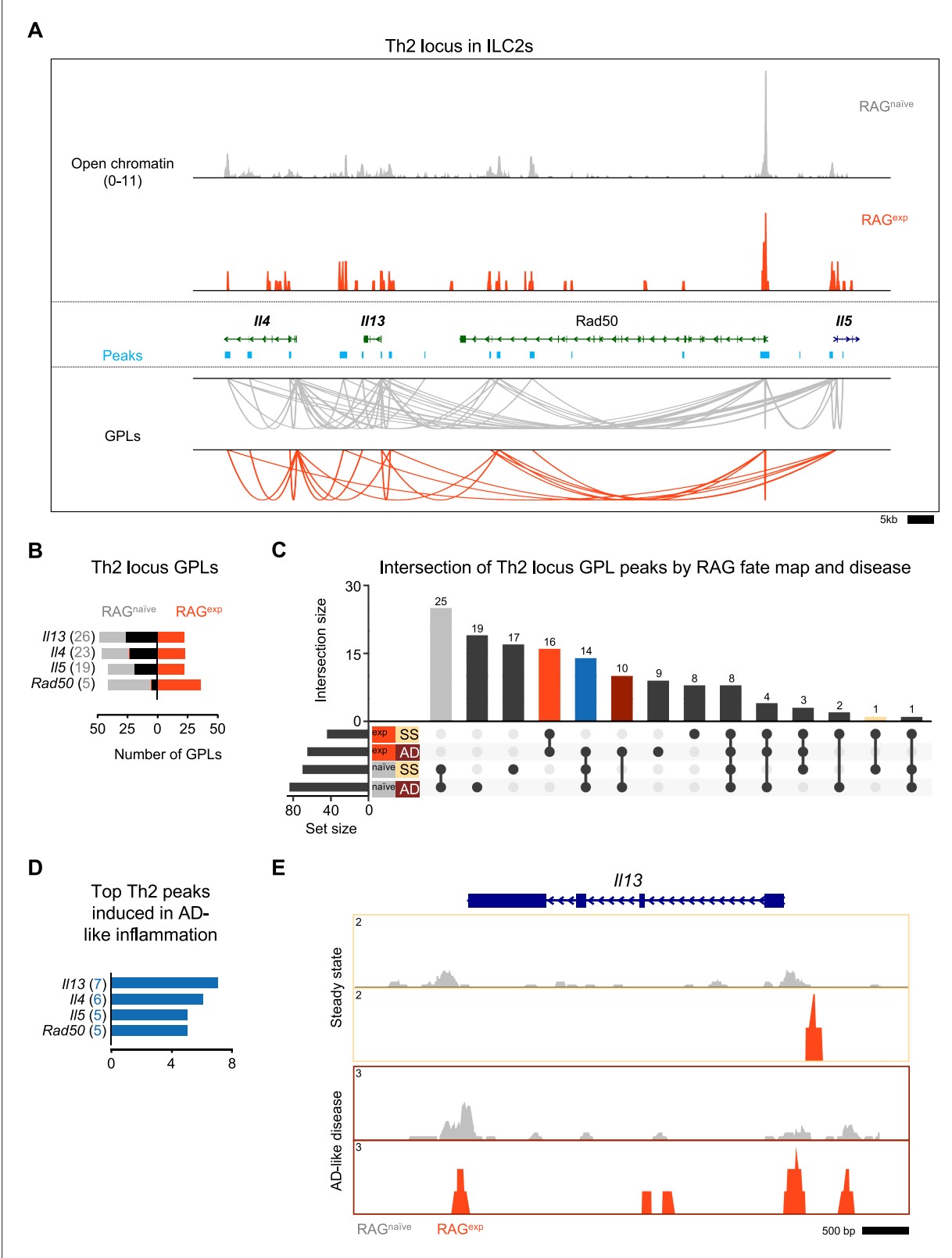

**Figure 7.** Recombinase activating gene (RAG) suppresses the Th2 locus. (**A**) Coverage plot of the Th2 genomic locus. Open chromatin in the group 2 innate lymphoid cell (ILC2) cluster for each *Rag1* fate-mapped state is shown on top, and corresponding peaks (teal) and gene-to-peaks links (GPLs) are shown below for the RAG^naïve sample (gray) and the RAG^exp sample (tomato red). Only GPLs that fit in the coverage window are shown. (**B**) All GPLs identified in each fate map state for the Th2 locus genes *Il4*, *Il13*, *Rad50*, and *Il5*. See *Supplementary file 1*, Table S15 for the full list of Th2 GPLs.

*Figure 7 continued on next page*

*Figure 7 continued*

The number of GPLs for each gene is shown on the left in gray for RAG^naïve and on the right in tomato red for RAG^exp. The difference is superimposed in black, and genes are sorted from more GPLs identified in RAG^naïve at the top to more links identified in RAG^exp at the bottom. (**C**) UpSet plot of intersections of peaks identified from Th2 locus GPLs (calculated as in *Figure 6A and B*) separated by both RAG fate map status (RAG^exp and RAG^naïve) and disease (SS - steady state, AD - atopic dermatitis-like inflammation). Each row represents one of the four sets of peaks, and each column corresponds to an intersection of one or more sets. See *Supplementary file 1*, Table S13 for the full list of peaks from GPLs for all genes, including Th2 genes, in each of the four sets. Columns identifying key intersections are color coded by the corresponding RAG fate map or disease groups. The blue column indicates the intersection of peaks from RAG^naïve cells and peaks induced in AD-like disease in RAG^exp cells. (**D**) Th2 genes are sorted by the number of AD-like inflammation-induced peaks. Peaks induced by AD-like disease were identified in corresponding GPLs, and genes were ranked by the frequency of links to induced peaks (representation in identified GPLs). See *Supplementary file 1*, Table S17 for the full list of ranked Th2 locus genes and associated GPLs. (**E**) Open chromatin tracks, split by disease (beige box – steady state; maroon box – AD-like disease) and by RAG fate map (RAG^naïve - gray, RAG^exp - red) for the *Il13* genomic locus.

The online version of this article includes the following figure supplement(s) for figure 7:

**Figure supplement 1.** Multiomic transcription factor analysis of the Th2 locus.

cells compared to the RAG^exp cells (*Figure 7E*). However, in the AD-like disease sample, the RAG^exp cells displayed increased open chromatin relative to the steady state, consistent with induction in the setting of inflammation, like our earlier findings for ILC2 genes such as *Ccr6* (*Figure 6E*). Taken together, our functional data and multiomic analyses demonstrate a role for RAG expression in modulating genes critical for ILC2 development and function, including the key type 2 cytokines expressed from the Th2 locus.

## Discussion

RAG recombinases evolved nearly 500 million years ago from endogenous transposons, crucially enabling antigen receptor rearrangement and the emergence of the adaptive immune cell lineages present in all modern vertebrates (*Liu et al., 2022*; *Zhang et al., 2019*). Indeed, RAG deficiency leads to a complete lack of B and T lymphocytes, manifesting clinically as severe combined immunodeficiency (SCID) (*Buckley, 2004*; *Mombaerts et al., 1992*; *Shinkai et al., 1992*). However, fate mapping studies have shown that multiple mature immune cell populations other than adaptive B and T lymphocytes have a history of RAG expression (*Yang et al., 2011*; *Karo et al., 2014*; *Welner et al., 2009*; *Igarashi et al., 2002*; *Pelayo et al., 2005*). More recent studies by Karo et al found that RAG expression during NK cell development influences multiple cellular functions including antitumor cytotoxicity, cell proliferation, and survival (*Karo et al., 2014*). Yet whether RAG modulates cellular functions of innate immune cell populations other than NK cells remains poorly understood. Here, using RAG-deficient mice, RAG fate mapping mice, and multiomic analyses, we report that RAG suppresses developmental and effector functions of ILC2s.

Our functional data in RAG-deficient mice demonstrate that populations of ILC2s producing the type 2 cytokines IL-5 and IL-13 preferentially expand in the absence of a history of RAG expression. This implies a specific role for RAG in the developmental repression of ILC2s. Building on this, our multiomic RAG fate mapping analyses of ILC2 gene programs demonstrate extensive epigenomic differences between RAG^exp and RAG^naïve cells. We found RAG-associated epigenomic suppression at multiple functional levels, including cell surface receptors, key transcription factors, and the Th2 locus encoding the type 2 cytokine genes *Il5* and *Il13*. Although RAG is only transiently expressed early in lymphoid development (*Wilson et al., 1994*), our data demonstrate that RAG expression can imprint durable effects on ILC2 gene programs to restrain their function.

Our observations imply that RAG expression may mark a developmentally distinct population of ILC2s. In adaptive lymphocytes, RAG expression in T cells is restricted to their time in the thymus. However, ILC2 populations have been observed in the thymus, provoking the hypothesis that thymic ILC2s may be uniquely high in an expression of RAG (*Ferreira et al., 2021*; *Gentek et al., 2013*). Prior studies by Schneider et al have identified ILC2 populations in adult tissues that variably derive from expansion of fetal, postnatal, and adult populations (*Schneider et al., 2019*). Yet how RAG expression in ILC2s may be restricted spatially or temporally remains unknown. The mouse strains used in the fate mapping studies by Schneider et al would be incompatible with our RAG fate mapping mice. Thus, novel mouse strains enabling intersectional genetics to trace ILC2 ontogeny (e.g. CreER/lox

for temporally restricted fate mapping and FlpO/frt for RAG fate mapping *Rodríguez et al., 2000*) are needed to more precisely determine when and where RAG expression occurs during ILC2 development. Beyond steady state ontogeny, our data suggest a history of RAG expression also imprints suppressed proliferative and type 2 inflammatory functions on ILC2s in the setting of AD-like disease.

It is increasingly recognized that the expression of effector molecules for both ILCs and their counterpart adaptive lymphocytes (e.g. IL-13 from ILC2s and Th2 cells) is governed by finely tuned transcriptomic and epigenomic regulomes (*Shih et al., 2016*; *Zhu et al., 2010*; *Koues et al., 2016*; *Agarwal and Rao, 1998*). ILCs tend to adopt these regulomes earlier in their development than T cells, and these 'poised' regulatory elements are thought to underlie the ability of tissue-resident ILCs to rapidly respond to stimuli. In contrast, the regulomes of naïve T cells remain relatively inactive until stimulation. Given that T cells are uniformly RAG-experienced, our data provoke the hypothesis that RAG$^{exp}$ ILC2s adopt a phenotype closer to that of naive T cells and may require stronger stimuli than RAG$^{naïve}$ ILC2s to become activated. Indeed, our analyses found that RAG-associated suppressive programs could be overcome in the setting of AD-like inflammation. Thus, sufficient RAG expression may mediate key events underlying the establishment and maintenance of functional regulomes not only in ILCs, but also T cells. How RAG might affect these changes, and whether they are independent of its enzymatic activity and/or antigen receptor recombination, remains to be elucidated.

Clinically, a link between enhanced type 2 immune activity and RAG dysfunction is well-established. Omenn Syndrome (OS) is a form of SCID characterized by exaggerated type 2 immune activation and typically arises in the setting of hypomorphic RAG gene mutations. Impaired antigen receptor rearrangement, with rare 'leaky' recombination events, leads to expansion of autoreactive oligoclonal T cells, eosinophilia, and markedly elevated IgE (*Omenn, 1965*; *Villa et al., 2008*; *Villa et al., 1998*; *Villa et al., 2001*; *Wada et al., 2005*). Similar phenotypes have been observed in mice harboring RAG mutations analogous to those found in human patients with OS (*Marrella et al., 2007*; *Khiong et al., 2007*). Notwithstanding these findings, the mechanisms underlying the propensity of oligoclonal T cells with hypomorphic RAG activity to preferentially develop into the Th2 subtype are unclear. Prior studies have found a role for regulatory T cells in controlling type 2 skewing of transferred T cells in RAG-deficient hosts, potentially explaining similar observations in patients with OS (*Milner et al., 2007*). Our data provide an additional mechanism by which RAG dysfunction may lead to OS through loss of cell-intrinsic RAG-mediated suppression of type 2 cellular programs. Additionally, increased type 2 cytokine production from RAG-deficient ILC2s may, in trans, enhance the expansion of the oligoclonal Th2 cell populations, IgE induction, and eosinophilia observed in RAG-deficient states like OS. However, whether other immune cell types with RAG dysfunction, such as ILCs, contribute to the pathogenesis of OS in humans has not been investigated.

Lymphoid acquisition of RAG activity may represent a newer evolutionary mechanism that fine-tunes ancient innate immune cell programs in addition to enabling the development of relatively newer antigen-specific adaptive immune cell populations. Independent of antigen receptor diversity, loss of this function may offer an explanation as to why oligoclonal T cells tend to expand and skew towards a Th2 cell phenotype in the setting of hypomorphic RAG function as in OS (*Milner et al., 2007*). Further studies are needed to define whether the suppressive effects of RAG expression operate similarly in T and B cells. Although we demonstrate that this phenomenon is observed in ILC2s, whether hypomorphic RAG expression in bona fide Th2 cells not only results in oligoclonality but also loss of suppression of the Th2 locus independently of antigen receptor rearrangement remains an outstanding question. Indeed, during the development of gene therapy strategies for RAG-deficient SCID, lower doses of wild-type RAG transgene expression have been associated with the development of OS-like conditions in transplanted RAG-deficient mice (*Pike-Overzet et al., 2011*; *Pike-Overzet et al., 2014*; *van Til et al., 2014a*; *van Til et al., 2014b*).

A major limitation of our study is a focus on cutaneous type 2 inflammation, which stemmed from our initial observations in the MC903 mouse model of AD-like disease. Furthermore, given the scarcity of skin-resident ILC2 populations, key functional investigations in our study such as cytokine production and multiomic sequencing were limited to the sdLN, as in prior studies (*Kim et al., 2013*; *Tamari et al., 2024*). However, ILC2s are recognized to have highly tissue-specific functions that extend much beyond inflammation to other processes including regeneration and metabolism. In addition to Il-5 and IL-13, ILC2s can produce other effector molecules such as acetylcholine, IL-9, methionine-enkephalin peptides, and amphiregulin, which modulate tissue responses across numerous organs (*Monticelli*

*et al., 2011*; *Brestoff et al., 2015*; *Monticelli et al., 2015*; *Mohapatra et al., 2016*; *Turner et al., 2013*; *Wang et al., 2017*; *Bando et al., 2020*; *Roberts et al., 2021*). Considering that the complexity of ILC2 biology may result in markedly divergent responses to RAG expression in other tissues and disease models, we thus restricted our initial studies to the skin, where we had strong molecular, cellular, and phenotypic outcomes. An implication of our findings in the skin is that RAG expression may modulate a variety of ILC2 functions in other tissues. Broader surveys of how RAG impacts ILC2 development and function in different tissues and disease states remain an exciting area of inquiry.

While we focused our multiomic analyses on ILC2s, it is likely that RAG may impact other ILC populations. For example, hyperactivation of intestinal ILC3s has been observed in *Rag1*$^{-/-}$ mice secondary to persistent phosphorylation of Signal Transducer and Activator of Transcription 3 (STAT3). Adoptive transfer of T regulatory cells rescued this phenotype, providing a cell-extrinsic mechanism for the observation of hyperactivated ILC3s in the setting of RAG deficiency (*Mao et al., 2018*). However, our data supporting a cell-intrinsic role for RAG in ILC2s may offer additional mechanistic insight into the prior observations in ILC3s. We found that the regulome of *Jak2*, which encodes JAK2, an upstream activator of STAT3, was more activated in RAG$^{naïve}$ ILC2s (*Figure 5C*). Additionally, the regulome for *Dusp1*, which encodes dual specificity phosphatase 1 (DUSP1), was more activated in RAG$^{exp}$ ILC2s (*Figure 5C*). While not implicated in directly dephosphorylating STAT3, a recent study found that DUSP1 overexpression negatively regulated the JAK2/STAT3 pathway (*Chen et al., 2023*). Notably, recent transcriptional profiling of skin ILCs identified a potential mechanism for skin ILC populations to transition to an ILC3-like phenotype (*Bielecki et al., 2021*), but how this process is regulated remains poorly understood. Taken together, our data provoke compelling new hypotheses about cell-intrinsic functions of RAG that may be complementary, rather than contradictory, to prior observations in gut and skin ILC populations. Additionally, our studies provide a rationale to design novel reagents to enable more comprehensive studies on the role of RAG in multiple innate immune cell populations across different tissues and disease models.

Our observations are also limited by the lack of a defined mechanism for how RAG expression imprints durable epigenomic and transcriptomic changes in ILC2s. The mechanisms by which RAG mediates VDJ recombination are well-defined, from the biochemical details of DNA-binding to the epigenomic accessibility of antigen receptor loci and timing of RAG expression (*Liu et al., 2022*; *Schatz and Swanson, 2011*; *Kuo and Schlissel, 2009*; *Desiderio, 2010*). Notwithstanding genomic stress (*Karo et al., 2014*) or potential RAG dose effects (*Pike-Overzet et al., 2011*; *Pike-Overzet et al., 2014*; *van Til et al., 2014a*; *van Til et al., 2014b*), how RAG expression might modulate broad developmental and functional lymphoid programs other than V(D)J recombination remains unclear. The RAG complex can bind both DNA and modified histones and has been observed to occupy thousands of sites across the genome (*Teng et al., 2015*). Thus, RAG may directly influence open chromatin states or obscure transcription factor binding sites to alter ILC2 development and function. Notably, RAG preferentially binds near transcription start sites of open chromatin in mouse thymocytes and pre-B cells, although corresponding effects on gene expression were not observed (*Teng et al., 2015*). Although canonical recombination sites are concentrated in the antigen receptor loci, cryptic recombination sites in other regions may be deleted or rearranged by RAG activity, altering the transcriptional regulation of associated genes (*Teng et al., 2015*). In contrast to developing B and T lymphocytes, the precise timing and location of RAG expression in ILC2s is not known. Thus, combined with the relative scarcity of ILC2s, conventional methods of chromatin immunoprecipitation to identify potential epigenomic regulatory mechanisms mediated by RAG expression may not be feasible in ILC2s or other rare cell populations. Instead, newer technologies such as self-reporting transposons (*Moudgil et al., 2020*) could be adapted to trace the genomic footprint of RAG in cells at various stages of development and in various tissues independent of the constraint of concurrent RAG expression. Finally, through its E3 ubiquitin ligase activity (*Liu et al., 2022*), RAG may influence immune signaling pathways independently of transcription altogether. Given that direct targeting of RAG would lead to unacceptable side effects, elucidating the mechanisms by which RAG imprints phenotypic changes beyond antigen receptor rearrangement is a critical next step in translating these findings to potential new therapies.

Our studies expand prior work implicating RAG in critical immune functions beyond antigen receptor rearrangement that is exclusive to adaptive lymphocytes. Furthermore, we provide additional insights into why patients with OS exhibit atopic syndromes in the setting of adaptive lymphocyte deficiency.

Future studies into mechanisms underlying these findings may lead to new therapeutic avenues for disorders such as atopic dermatitis, food allergy, and asthma.

# Materials and methods

## Animal studies

Wild-type (WT) C57BL/6J and WT congenic strains (CD90.1[+], CD45.1[+]), *Rag1*[-/-], and *Rag2*[-/-] mice were initially purchased from the Jackson Laboratory and bred in-house. The RAG fate-mapping strain *Rag1*[Cre]::*Rosa26*[LSL-tdRFP] was originally created in the lab of Paul Kincade (*Welner et al., 2009*) and bred in-house. All mice were housed in specific-pathogen-free conditions in an environmentally controlled animal faculty with a 12 hr light-dark cycle and given unrestricted access to food and water at Icahn School of Medicine at Mount Sinai or Washington University School of Medicine in St. Louis. All animal protocols and experiments were approved by the Institutional Animal Care and Use Committee (IACUC) at Icahn School of Medicine at Mount Sinai (protocol number 202200000163) or Washington University School of Medicine in St. Louis (protocol numbers 20–0017 and 23–0130).

Experiments were performed on independent cohorts of male and female mice. The sample size for animal experiments was chosen based on previous data generated in the laboratory. For induction of AD-like disease, 8- to 12-wk-old mice were treated with 2 nmol calcipotriol (MC903, Tocris Bioscience) in 10 µL of 100% ethanol (EtOH) vehicle, or vehicle alone, on the bilateral ear skin daily for 7–10 days. Body weight and ear thickness were measured daily with a digital scale and analog caliper by the same investigator. For tissue harvest, animals were euthanized by $CO_2$ inhalation.

## Flow cytometry

Cervical skin draining lymph nodes (sdLN) were removed from the mice and immediately homogenized manually through a 100 µm cell strainer (Fisher Scientific) into a 50 mL tube with the end of a plunger from a 3 mL syringe. The strainer was washed with wash medium (2% vol/vol FBS/PBS) and the strained cells were centrifuged at 400 g for 5 min at 4 °C. Lymph node cell samples were stained with Zombie NIR viability dye (Biolegend; 1:500) to exclude dead cells, followed by Fc-receptor blocking and cell-surface staining with specific antibodies. The cells were analyzed using either LSR Fortessa (BD) or Cytek Aurora (CYTEK) flow cytometers. Data was obtained using either FACSDiva (BD) or SpectroFlo (CYTEK) software and was further analyzed using FlowJo.

## Lymphocyte stimulations

After tissue harvest, ILC stimulations were performed by incubating 0.5–1×10[6] cells for 4 hr at 37 °C in stimulation media (DMEM with 5% fetal bovine serum, 1% penicillin/streptomycin, 2 mM L-glutamine, 50 ng/mL Phorbol 12-myristate 13-acetate (PMA), 100 ng/mL ionomycin, 5 ug/mL Brefeldin A (BFA), 2 uM monensin). After stimulation, cells were washed in wash medium, fixed, and stained for surface and intracellular markers as described for unstimulated cells.

## Splenocyte chimeras

Spleens were harvested from donor WT B6 mice and immediately homogenized manually through a 100 µm cell strainer (Fisher Scientific) into a 50 mL tube with the end of a plunger from a 3 mL syringe. The strainer was washed with wash medium (2% vol/vol FBS/PBS) and the strained cells were centrifuged at 400 g for 5 min at 4 °C followed by treatment with RBC lysis buffer for 2 min and two wash steps using two volumes of wash medium. Cells were counted, and 5 million splenocytes were injected intraperitoneally into each recipient mouse. Experiments were performed 4 wk following splenocyte add-back to allow immune reconstitution.

## Bone marrow chimeras

Recipient mice were provided with antibiotic water, consisting of 5 mL of Sulfatrim (sulfamethoxazole/ trimethoprim) added into 200 mL of drinking water, for 1 wk starting from the day prior to irradiation (day –1). On day 0, recipient mice were irradiated with 950 cGy using the X-RAD 320 (Precision X-Ray). BM was harvested from donor mice femurs and tibias and treated with RBC lysis buffer (Sigma-Aldrich) for 2 min. BM cells were transferred into a 15 mL conical tube through a 70 µm cell strainer (Fisher Scientific) and the cell strainer and cells were washed with 2% (vol/vol) FBS/PBS. The concentration

of living cells was determined using a Cellometer Auto 2000 (Nexcelom Bioscience) with ViaStain AOPI Staining Solution (Nexcelom Bioscience). Recipient mice received the same number of cells, at $1 \times 10^7$ live bone marrow cells per mouse, through retroorbital injection within 24 hr after irradiation. Recipients were given 8 wk for immune reconstitution after BM transplantation before experimental use.

## Cryopreserving sdLN cells for sequencing

*Rag1*[Cre]*::Rosa26*[LSL-tdRFP] mice were treated with 2 nmol calcipotriol (MC903, Tocris Bioscience) in 10 μL of 100% ethanol (EtOH) vehicle, or vehicle alone, on the bilateral ear skin daily for 7 days to induce AD-like inflammation. The next day, cervical sdLN were harvested and immediately homogenized manually through a 100 μm cell strainer (Fisher Scientific) into a 50 mL tube with the end of a plunger from a 3 mL syringe. The strainer was washed with wash medium (2% vol/vol FBS/PBS) and the strained cells were centrifuged at 400 g for 5 min at 4 °C. Next, cells were incubated with biotinylated antibodies (anti-mouse CD3e, CD19, CD11b; 1:300; Biolegend) in 100 μL of wash buffer for 20 min at 4 °C, followed by two washes in two volumes of wash buffer. Next, no more than $10^7$ cells were incubated with Streptavidin MicroBeads (Miltenyi) in 500 μL separation buffer (0.5% w/v BSA in PBS; BSA and PBS from Sigma) at 4 °C for 20 min, then added to LD columns (Miltenyi) pre-equilibrated with separation buffer and loaded in a QuadroMACS Separator (Miltenyi) for negative cell selection. Remaining cells were eluted in 1 mL separation buffer and cells were centrifuged at 400 g for 5 min at 4 °C, followed by resuspension in freezing buffer (10% DMSO, Invitrogen; 20% FBS in DMEM, Sigma) and slow freezing to –80 °C in a CoolCell LX (Corning) device.

## Processing cryopreserved cells for multiome

Cryopreserved sdLN cells were processed as recommended by the 10X Genomics Demonstrated-Protocol_NucleiIsolation_ATAC_GEX_Sequencing_RevC_(CG000365) instructions for primary cells without any modification to the protocol. Briefly, cells were thawed in a 37 °C water bath followed by dilution into media (RPMI +15% FBS, Sigma) and centrifugation at 400 g for 5 min at 4 °C. For each final sample (EtOH vehicle- or MC903-treated), cells were pooled from samples from three individual mice. Cells were resuspended in PBS +0.04% BSA (Sigma) and passed through a 40 μm Flomi strainer (Bel-art) followed by determination of cell concentration using the using Cellometer Auto 2000 (Nexcelom Bioscience) with ViaStain AOPI Staining Solution (Nexcelom Bioscience). Cells were centrifuged for 5 min at 4 °C and the supernatant was removed. Lysis Buffer (Tris HCl base with 0.1% Tween-20, 0.1% NP-40, 0.01% digitonin, 1 mM DTT, and 1 U/μL Protectors RNase inhibitor, Sigma; full recipe in 10X Genomics protocol) was added, cells mixed by pipetting 10 times, and incubated on ice for 3 min. Nuclei from lysed cells were centrifuged at 400 g for 5 min at 4 °C and washed in 1 mL Wash Buffer (Lysis Buffer, but without NP-40 or digitonin). The wash step was repeated two more times. Nuclei concentration was determined as for cell concentration using the Cellometer and ViaStain solution. The AOPI staining indicated 97–99% lysis efficiency of the cells. We manually confirmed nuclei count using a Bright-Line hemacytometer (Hausser Scientific). Nuclei were centrifuged at 400 g for 5 min and resuspended in a volume of 1 X Nuclei Buffer (10 X Genomics) to yield roughly 4000 nuclei/μL. We then immediately proceeded to the 10X Chromium Next GEM Single Cell Multiome ATAC +Gene Expression pipeline.

## Multiome library construction and sequencing

Multiome 3v3.1 GEX and ATAC libraries were prepared as recommended by 10X Genomics protocol Chromium_NextGEM_Multiome_ATAC_GEX_User_Guide_RevD (CG000338). For sample preparation on the 10X Genomics platform, the Chromium Next GEM Single Cell Multiome ATAC + Gene Expression Reagent Bundle, 16 rxns PN-1000283, Chromium Next GEM Chip J Single Cell Kit, 48 rxns PN-1000234, Single Index Kit N Set A, 96 rxns PN-1000212 (ATAC), Dual Index Kit TT Set A, 96 rxns PN-1000215 (3v3.1 GEX), were used. The concentration of each library was accurately determined through qPCR utilizing the KAPA library Quantification Kit according to the manufacturer's protocol (KAPA Biosystems/Roche) to produce cluster counts appropriate for the Illumina NovaSeq6000 instrument. GEX libraries were pooled and run over 0.05 of a NovaSeq6000 S4 flow cell using the XP workflow and running a 28×10×10 ×150 sequencing recipe in accordance with the manufacturer's protocol. Target coverage was 500 M reads per sample. ATAC libraries were pooled and run over

0.167 of a NovaSeq6000 S1 flow cell using the XP workflow and running a 51×8×16×51 sequencing recipe in accordance with the manufacturer's protocol. Target coverage was 250 M reads per sample.

## Multiomic data analysis

The cellranger-arc-2.0.0 (10X Genomics) pipeline was used to generate FASTQ files, gene expression matrices, and ATAC fragment tables for each sample, followed by aggregation using the aggr function. Default settings were utilized, with the exception that we incorporated a custom reference with the sequence for tdRFP (see *Supplementary file 1*, tdRFP sequence) added to the default mouse reference sequence provided by cellranger (refdata-cellranger-arc-mm10-2020-A-2.0.0). Correction for ambient RNA was performed using SoupX (*Young and Behjati, 2020*) with clustering information provided by the default cellranger outputs. Doublets were removed using Scrublet (*Wolock et al., 2019*) with default settings.

Corrected data was then processed using Signac (*Stuart et al., 2021*) and Seurat (*Stuart et al., 2019*; *Hao et al., 2021*; *Butler et al., 2018*). ATAC-seq peaks were identified using MACS2 (*Zhang et al., 2008*) through the CallPeaks function in Signac. Per-cell quality control metrics were computed using the TSSEnrichment and NucleosomeSignal functions, and cells retained with a nucleosome signal score <1.5, TSS enrichment score >1, total RNA counts <15,000 and>1000, total ATAC counts <75,000 and>100, percent mitochondrial reads <5%, and percent ribosomal genes detected <10%. After these filtering steps, 10,304 cells remained. Cells were further filtered by their expression of lineage-defining markers similar to the negative selection step during sample processing. Cells with detectable transcripts for *Cd3d*, *Cd3e*, *Cd3g*, *Cd4*, *Cd19*, *Cd8a*, and *Itgam* were removed. This left 2034 remaining cells for further analysis.

The SCTransform function of Seurat was used to normalize RNA counts. We performed integration of the two samples using the RNA assay to correct for batch effects and treatment in the initial clustering using the default parameters for the functions SelectIntegrationFeatures, FindIntegrationAnchors, and IntegrateData. The integrated data was used for PCA (25 dimensions) and UMAP reduction for the RNA assay alone. With default parameters in Signac, we used TFIDF to normalize ATAC peaks and latent semantic indexing (LSI) to reduce the dimensionality of the ATAC data. We constructed a UMAP of the ATAC data alone using the LSI reduction (dimensions 2–25). To construct a joint graph and UMAP using equal weighting from the RNA and ATAC assays, we used the FindMultiModalNeighbors function of Seurat/Signac using default parameters (RNA dimensions 1–25, ATAC dimensions 2–25). We used a resolution of 0.1 to identify clusters with the FindClusters function in Seurat/Signac. Cell types were assigned based on the manual curation of marker genes. Initially, seven clusters were identified, but two highly similar lymphocyte clusters were merged for a total of 6 cell types.

The inferred Gene Activity (GA) assay from the ATAC-seq data was calculated using default parameters of the GeneActivity function in Signac. FindAllMarkers was used to identify top markers by cluster for both RNA gene expression data (GEX) and GA, with setting adjustments including min.pct=0.20 and logfc.threshold=0.25. The differentially accessible (DA) open chromatin assay was calculated in Signac with the FindMarkers function on the ATAC-seq peaks assay (called using MACS2 as above). The differential test used was 'LR' (logistical regression, as suggested for snRNA-seq *Ntranos et al., 2019*). The total number of ATAC fragments was used as a latent variable to mitigate the effect of differential sequencing depth. Given the sparsity of the data, the min.pct parameter was set to 0.02. After identifying the top differentially accessible peaks for each cluster, the gene closest to each peak was determined using the ClosestFeature function in Signac. Results were filtered for genes within $10^5$ base pairs of the corresponding peak. The filtered gene lists were used for the 'DA' assay as markers of each cluster (top 25) and an expanded list for the ILC2 cluster (top 100). Venn diagrams were calculated using BioVenn/BioVennR (*Hulsen et al., 2008*).

Gene set enrichment analysis was performed and visualized using ClusterProfiler (*Wu et al., 2021*). For GSEA on steady-state ILC2 DEGs between fate mapped states, we opted to use more permissive filtering parameters instead of default parameters. We created the ranked list of DEGs using the FindMarkers function in Seurat with min.pct=0.1 and logfc.threshold=0.1. The DEG list from the GEX assay was used to generate the GSEA results. The DEG list from the GA assay did not yield any significant GSEA results. The ClusterProfiler function gseGO was used to analyze the ranked DEG list using the parameters minGSSize = 50, maxGSSize = 500, p-valueCutoff = 0.05.

A motif matrix was constructed from the ATAC data Granges using the 'CORE' collection and 'vertebrates' taxonomy group from the JASPAR2022 position weight matrix set and the mm10 reference genome. Per cell transcription factor motif activity was calculated with chromVar (*Schep et al., 2017*) using the motif matrix and MACS2 called peaks. Transcription factor motifs were identified in differentially accessible chromatin using the FindMotifs function in Signac.

The correlation coefficients, or gene-to-peak links (GPLs), between gene expression and accessibility of each peak, were calculated for all peaks within $10^6$ base pairs of the transcription start sites for all detected genes using the LinkPeaks function of Signac with min.cells=2. GPLs were filtered by the gene for the curated ILC2 and Th2 gene sets. Since multiple genes can be linked to one peak by GPL analysis, finding intersections of GPLs in set analysis would result in counting some epigenomic regions multiple times. Thus, for set analysis, we eliminated GPLs with redundant peaks. Then, we used each list of non-redundant peaks as input sets to generate UpSet plots and lists of intersecting peaks between states (RAG1 fate map positive or negative; AD-like disease or steady state) using UpSetR (*Conway et al., 2017*). Coverage plots of the single cell multiomic data, including open chromatin, peaks, and links (GPLs), were plotted using the CoveragePlot function in Signac.

## Acknowledgements

We thank all members of the Kim lab for their helpful comments and discussion. This work is supported by the Allen Discovery Center program, a Paul G Allen Frontiers Group advised program of the Paul G Allen Family Foundation, the Doris Duke Charitable Foundation, LEO Pharma, the National Institute of Arthritis and Musculoskeletal and Skin Diseases (NIAMS) (AR070116, AR077007), and the National Institute of Allergy and Infectious Disease (NIAID) (AI167933 and AI167047) of the National Institutes of Health (NIH). AMV is supported by NIAMS (1K08AR080219). MT is supported by the Japanese Society of Allergology (JSA) International Scholarship. AMT is supported by the NIAID (AI007163 and AI154912). We thank the Genome Technology Access Center at the McDonnell Genome Institute at Washington University School of Medicine for help with genomic analysis. The Center is partially supported by NCI Cancer Center Support Grant #P30 CA91842 to the Siteman Cancer Center and by ICTS/CTSA Grant# UL1TR002345 from the National Center for Research Resources (NCRR), a component of the National Institutes of Health (NIH), and NIH Roadmap for Medical Research. This publication is solely the responsibility of the authors and does not necessarily represent the official view of NCRR or NIH.

## Additional information

### Competing interests

Aaron M Ver Heul: has contributed to scientific advisory boards at Galderma, Novartis, and Sanofi-Regeneron and has performed sponsored research for Amgen and Celldex. Madison Mack: is affiliated with Sanofi. Brian S Kim: Reviewing editor, eLife. The other authors declare that no competing interests exist.

### Funding

| Funder | Grant reference number | Author |
| --- | --- | --- |
| National Institute of Arthritis and Musculoskeletal and Skin Diseases | AR080219 | Aaron M Ver Heul |
| National Institute of Arthritis and Musculoskeletal and Skin Diseases | AR070116 | Brian S Kim |
| National Institute of Arthritis and Musculoskeletal and Skin Diseases | AR077007 | Brian S Kim |

| Funder | Grant reference number | Author |
|---|---|---|
| National Institute of Allergy and Infectious Diseases | AI167933 | Brian S Kim |
| National Institute of Allergy and Infectious Diseases | AI167047 | Brian S Kim |
| National Institute of Allergy and Infectious Diseases | AI007163 | Anna M Trier |
| National Institute of Allergy and Infectious Diseases | AI154912 | Anna M Trier |
| Japanese Society of Allergology | International Scholarship | Masato Tamari |
| Allen Discovery Center | | Brian S Kim |
| Doris Duke Charitable Foundation | | Brian S Kim |
| LEO Pharma | | Brian S Kim |

The funders had no role in study design, data collection and interpretation, or the decision to submit the work for publication.

## Author contributions

Aaron M Ver Heul, Conceptualization, Data curation, Formal analysis, Validation, Investigation, Visualization, Methodology, Writing – original draft, Writing – review and editing; Madison Mack, Conceptualization, Data curation, Formal analysis, Validation, Investigation, Methodology, Writing – review and editing; Lydia Zamidar, Masato Tamari, Ting-Lin Yang, Conceptualization, Validation, Investigation, Methodology, Writing – review and editing; Anna M Trier, Do-Hyun Kim, Hannah Janzen-Meza, Investigation, Methodology, Writing – review and editing; Steven J Van Dyken, Chyi-Song Hsieh, Joseph C Sun, Conceptualization, Resources, Writing – review and editing; Jenny M Karo, Conceptualization, Writing – review and editing; Brian S Kim, Conceptualization, Supervision, Funding acquisition, Methodology, Project administration, Writing – review and editing

## Author ORCIDs

Aaron M Ver Heul ⓘ https://orcid.org/0000-0003-3386-8351
Ting-Lin Yang ⓘ https://orcid.org/0000-0002-4808-7665
Brian S Kim ⓘ https://orcid.org/0000-0002-8100-7161

## Ethics

All animal protocols and experiments were approved by the Institutional Animal Care and Use Committee (IACUC) at Icahn School of Medicine at Mount Sinai (202200000163) or Washington University School of Medicine in St. Louis (protocol numbers 20-0017 and 23-0130).

Reviewer #1 (Public Review): https://doi.org/10.7554/eLife.98287.3.sa1
Reviewer #2 (Public Review): https://doi.org/10.7554/eLife.98287.3.sa2
Author response https://doi.org/10.7554/eLife.98287.3.sa3

# Additional files

## Supplementary files

MDAR checklist

Supplementary file 1. Sequence file of the tandem dimer red fluorescent protein (tdRFP) transgene present in the Rosa26-LSL-tdRFP reporter mouse in FASTA format. This sequence was appended to the mm10 mouse reference genome to enable the detection and annotation of tdRFP transcripts in the scRNA-sequencing data. Tables S01-S17. See text for details of table contents.

## Data availability

Sequencing data have been deposited at the Gene Expression Omnibus (GEO) under accession number GSE192597. All processed data reported in this paper are in the aggregated data supplied in the GEO supplementary file or in *Supplementary file 1*. No new code was generated in this study.

The following dataset was generated:

| Author(s) | Year | Dataset title | Dataset URL | Database and Identifier |
|-----------|------|---------------|-------------|-------------------------|
| Ver Heul AM, Zamidar L, Yang T, Kim BS | 2024 | Single nuclei RNA sequencing of skin draining lymph nodes in the setting of skin inflammation and RAG1 fate mapping | https://www.ncbi. nlm.nih.gov/geo/ query/acc.cgi?acc= GSE192597 | NCBI Gene Expression Omnibus, GSE192597 |

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

# Appendix 1

## Appendix 1—key resources table

| Reagent type (species) or resource | Designation | Source or reference | Identifiers | Additional information |
|---|---|---|---|---|
| Antibody | anti-mouse CD11b | eBioscience | Cat#: 45-0112-82; RRID:AB_953558 | FC (1:300) |
| Antibody | anti-mouse CD11c | eBioscience | Cat#: 45-0114-82; RRID:AB_925727 | FC (1:300) |
| Antibody | anti-mouse CD19 | eBioscience | Cat#: 45-0193-82; RRID:AB_1106999 | FC (1:300) |
| Antibody | anti-mouse CD3e | eBioscience | Cat#: 45-0031-82; RRID:AB_1107000 | FC (1:300) |
| Antibody | anti-mouse CD5 | eBioscience | Cat#: 45-0051-82; RRID:AB_914334 | FC (1:300) |
| Antibody | anti-mouse NK1.1 | eBioscience | Cat#: 45-5941-82; RRID:AB_914361 | FC (1:300) |
| Antibody | anti-mouse FcεR1α | eBioscience | Cat#: 11-5898-82; RRID:AB_465308 | FC (1:300) |
| Antibody | anti-mouse IL-13 | eBioscience | Cat#: 48-7136-42; RRID:AB_2784729 | FC (1:300) |
| Antibody | anti-mouse CD45.1 | BioLegend | Cat#: 110706; RRID:AB_313495 | FC (1:300) |
| Antibody | anti-mouse CD45.2 | BioLegend | Cat#: 109808; RRID:AB_313445 | FC (1:300) |
| Antibody | anti-mouse ICOS | BioLegend | Cat#: 313506; RRID:AB_416330 | FC (1:300) |
| Antibody | anti-mouse CD62L | BioLegend | Cat#: 104428; RRID:AB_830799 | FC (1:300) |
| Antibody | anti-mouse CD69 | BioLegend | Cat#: 104530; RRID:AB_2563062 | FC (1:300) |
| Antibody | anti-mouse KLRG1 | BioLegend | Cat#: 138424; RRID:AB_2564051 | FC (1:300) |
| Antibody | anti-mouse IL-5 | BioLegend | Cat#: 504306; RRID:AB_315330 | FC (1:300) |
| Antibody | anti-mouse IL-13 | BioLegend | Cat#: 503826; RRID:AB_2650897 | FC (1:300) |
| Antibody | anti-mouse IL-17A | BioLegend | Cat#: 506927; RRID:AB_11126144 | FC (1:300) |
| Antibody | anti-mouse IFNγ | BioLegend | Cat#: 505806; RRID:AB_315400 | FC (1:300) |
| Antibody | anti-mouse CD90.2 | BioLegend | Cat#: 105328; RRID:AB_10613293 | FC (1:300) |
| Antibody | anti-mouse IL-33Rα | BioLegend | Cat#: 145308; RRID:AB_2565569 | FC (1:150) |
| Antibody | anti-mouse IL-33Rα | BioLegend | Cat#: 145327; RRID:AB_2565569 | FC (1:150) |
| Antibody | anti-mouse CD8a | BioLegend | Cat#: 100762; RRID:AB_2564027 | FC (1:300) |
| Antibody | anti-mouse CD25 | BioLegend | Cat#: 102016; RRID:AB_312865 | FC (1:300) |
| Antibody | anti-mouse CD45.2 | BioLegend | Cat#: 109806; RRID:AB_313443 | FC (1:300) |
| Antibody | anti-mouse CD90.2 | BioLegend | Cat#: 109830; RRID:AB_1186098 | FC (1:300) |
| Antibody | anti-mouse CD4 | BioLegend | Cat#: 100449; RRID:AB_2564587 | FC (1:300) |
| Antibody | anti-mouse Gata3 | BioLegend | Cat#: 653814; RRID:AB_2563221 | FC (1:300) |
| Antibody | anti-mouse CD25 | BioLegend | Cat#: 102036; RRID:AB_2563059 | FC (1:300) |
| Antibody | anti-mouse CD90.1 | BioLegend | Cat#: 202537; RRID:AB_2562644 | FC (1:300) |
| Antibody | anti-mouse CD3e | BioLegend | Cat#: 155608; RRID:AB_2750434 | FC (1:300) |
| Antibody | anti-mouse CD3e | BioLegend | Cat#: 100339; RRID:AB_11150783 | FC (1:300) |
| Antibody | anti-mouse CD28 | BioLegend | Cat#: 102115; RRID:AB_11150408 | FC (1:300) |
| Antibody | anti-mouse CD3 | BioLegend | Cat#: 100243; RRID:AB_2563946 | FC (1:300) |
| Antibody | anti-mouse CD19 | BioLegend | Cat#: 115503; RRID:AB_313638 | FC (1:300) |
| Antibody | anti-mouse CD11b | BioLegend | Cat#: 101203; RRID:AB_312786 | FC (1:300) |
| Antibody | anti-mouse IL-4 | BioLegend | Cat#: 504109; RRID:AB_493320 | FC (1:300) |
| Antibody | anti-mouse CD117 (c-Kit) | BioLegend | Cat#: 105838; RRID:AB_2616739 | FC (1:300) |

*Appendix 1 Continued on next page*

*Appendix 1 Continued*

| Reagent type (species) or resource | Designation | Source or reference | Identifiers | Additional information |
|---|---|---|---|---|
| Antibody | anti-mouse CD49b | Invitrogen | Cat#: 17-5971-82; RRID:AB_469485 | FC (1:300) |
| Antibody | anti-mouse CD45R/B220 | BioLegend | Cat#: 103275; RRID:AB_2860602 | FC (1:300) |
| Antibody | anti-mouse I-A/I-E (MHCII) | BioLegend | Cat#: 107622; RRID:AB_493727 | FC (1:300) |
| Antibody | anti-mouse Ly-6A/E (Sca-1) | BioLegend | Cat#: 122512; RRID:AB_756197 | FC (1:300) |
| Antibody | anti-mouse F4/80 | BioLegend | Cat#: 123112; RRID:AB_893482 | FC (1:300) |
| Antibody | anti-mouse SiglecF | BD Biosciences | Cat#: 562757; RRID:AB_2687994 | FC (1:300) |
| Antibody | anti-mouse TCR γ/δ | eBioscience | Cat#: 48-5711-82; RRID:AB_2574071 | FC (1:300) |
| Antibody | anti-mouse Ly-6G | eBioscience | Cat#: 62-9668-82; RRID:AB_2762763 | FC (1:300) |
| Antibody | anti-mouse CD16/CD32 | Bio X Cell | Cat#: BE0307; RRID:AB_1107647 | FC (1:300) |
| Antibody | streptavidin | BioLegend | Cat#: 405204 | FC (1:300) |
| Antibody | streptavidin | BioLegend | Cat#: 405207 | FC (1:300) |
| Chemical compound, drug | Phorbol 12-myristate 13-acetate (PMA) | Sigma | P1585 | |
| Chemical compound, drug | Ionomycin | Sigma | I0634 | |
| Chemical compound, drug | Monensin | Biolegend | 420701 | |
| Chemical compound, drug | Brefeldin A Solution | Biolegend | 420601 | |
| Chemical compound, drug | Calcipotriol (MC903) | Tocris Biosciences | 2700 | |
| Chemical compound, drug | DMSO | Invitrogen | D12345 | |
| Chemical compound, drug | Nuclei Buffer (20 X) | 10 x Genomics | 2000153/2000207 | |
| Chemical compound, drug | Digitonin | ThermoFisher | BN2006 | |
| Chemical compound, drug | Nonidet P40 Substitute | Sigma | 74385 | |
| Chemical compound, drug | Protector RNase inhibitor | Sigma | 3335402001 | |
| Chemical compound, drug | Tween 20 | Bio-Rad | 1662404 | |
| Other | ZombieNIR | Biolegend | 423106 | Viability stain (1:500) |
| Other | ZombieUV | Biolegend | 423107 | Viability stain (1:500) |

*Appendix 1 Continued on next page*

*Appendix 1 Continued*

| Reagent type (species) or resource | Designation | Source or reference | Identifiers | Additional information |
|---|---|---|---|---|
| Strain, strain background (*Mus musculus*, C57BL/6) | B6 WT | Jackson Laboratory | C57BL/6J; Cat# 000664 | |
| Strain, strain background (*M. musculus*, C57BL/6) | *Rag1*-/- | Jackson Laboratory | B6.129S7-Rag1tm1Mom/J; Cat# 002216 | |
| Strain, strain background (*M. musculus*, C57BL/6) | *Rag2*-/- | Jackson Laboratory | B6.Cg-Rag2tm1.1Cgn/J; Cat# 002014 | |
| Strain, strain background (*M. musculus*, C57BL/6) | B6 CD45.1 | Jackson Laboratory | B6.SJL-Ptprca Pepcb/BoyJ; Cat# 008450 | |
| Strain, strain background (*M. musculus*, C57BL/6) | B6 CD90.1 | Jackson Laboratory | B6.PL-Thy1a/CyJ; Cat# 000406 | |
| Strain, strain background (*M. musculus*, C57BL/6) | *Rag1*-Cre x *Rosa26*-tdRFP fate-map mice | Joseph Sun (Weill Cornell Medical College) | *Rag1*-Cre - MGI:3584018 *Rosa26*-tdRFP - MGI:3696099 | Published references *Karo et al., 2014*; *Welner et al., 2009* |
| Software, algorithm | BD FACSDiva (v8.0) | BD Life Sciences | RRID:SCR_001456 | https://www.bdbiosciences.com/en-us/products/software/instrument-software/bd-facsdiva-software |
| Software, algorithm | FlowJo (v10.8) | BD Life Sciences | RRID:SCR_008520 | https://www.flowjo.com/ |
| Software, algorithm | SpectroFlo (v3.0) | CYTEK | RRID:SCR_025494 | https://cytekbio.com/pages/spectro-flo |
| Software, algorithm | Prism 9 | GraphPad Software | RRID:SCR_002798 | https://www.graphpad.com/scientific-software/prism/ |
| Software, algorithm | R (v4.2.2) | R core | RRID:SCR_001905 | https://www.r-project.org/ |
| Software, algorithm | Seurat (v4.2.0) | Seurat *Hao et al., 2021* | RRID:SCR_007322 | https://github.com/satijalab/seurat |
| Software, algorithm | Signac (v1.8.0) | Signac *Stuart et al., 2021* | RRID:SCR_021158 | https://github.com/stuart-lab/signac |
| Software, algorithm | chromVAR (v1.18.0) | chromVAR *Schep et al., 2017* | RRID:SCR_026570 | https://github.com/GreenleafLab/chromVAR |
| Software, algorithm | JASPAR2022 (v0.99.7) | JASPAR2022 | RRID:SCR_003030 | https://github.com/da-bar/JASPAR2022 |
| Software, algorithm | SoupX (v1.6.1) | SoupX *Young and Behjati, 2020* | RRID:SCR_019193 | https://github.com/constantAmateur/SoupX |
| Software, algorithm | clusterProfiler (v4.4.4) | clusterProfiler *Wu et al., 2021* | RRID:SCR_016884 | https://github.com/YuLab-SMU/clusterProfiler |
| Software, algorithm | singleCellTK (v2.6.0) | singleCellTK *Wang et al., 2022* | RRID:SCR_026813 | https://github.com/compbiomed/singleCellTK |
| Software, algorithm | biomaRt (v2.52.0) | biomaRt *Durinck et al., 2009* | RRID:SCR_019214 | https://github.com/grimbough/biomaRt |

*Appendix 1 Continued*

| Reagent type (species) or resource | Designation | Source or reference | Identifiers | Additional information |
|---|---|---|---|---|
| Software, algorithm | EnsDb.Mmusculus. v79 (v2.99.0) | EnsDb. Mmusculus.v79 *Rainer, 2017* | RRID:SCR_002344 | https://bioconductor.org/ packages/release/data/ annotation/html/EnsDb. Mmusculus.v79.html |
| Software, algorithm | BSgenome. Mmusculus. UCSC.mm10 (v1.4.3) | BSgenome. Mmusculus. UCSC.mm10 *Team, 2021* | RRID:SCR_024230 | https://bioconductor.org/ packages/release/data/ annotation/html/BSgenome. Mmusculus.UCSC.mm10.html |
| Software, algorithm | ggplot2 (v3.3.6) | ggplot2 *Wickham, 2016* | RRID:SCR_014601 | https://github.com/tidyverse/ ggplot2 |
| Software, algorithm | viridis (v0.6.2) | viridis *Garnier et al., 2021* | RRID:SCR_016696 | https://github.com/ sjmgarnier/viridis |
| Software, algorithm | TFBSTools (v1.34.0) | TFBSTools *Tan and Lenhard, 2016* | RRID:SCR_024260 | https://bioconductor.org/ packages/release/bioc/html/ TFBSTools.html |
| Software, algorithm | motifmatchr (v1.18.0) | motifmatchr *Schep, 2022* | RRID:SCR_026739 | https://github.com/ GreenleafLab/motifmatchr |
| Software, algorithm | BioVenn/BioVennR | BioVenn *Hulsen et al., 2008* | RRID:SCR_026853 | https://www.biovenn.nl/ |
| Software, algorithm | UpSetR | UpSetR *Conway et al., 2017* | RRID:SCR_026112 | http://gehlenborglab.org/ research/projects/upsetr/ |
| Software, algorithm | Cell Ranger ARC | Cell Ranger ARC *Zheng et al., 2017*; *Satpathy et al., 2019* | RRID:SCR_023897 | https://support.10xgenomics. com/single-cell-multiome- atac-gex/software/overview/ welcome |

