## [Editor Report · eLife Assessment]

This study provides new insights into the expression profile of ILCs that demonstrate a history of RAG expression. It examines in part the potential intrinsic regulation of RAG expression and seeks to understand how the epigenetic state of ILCs is established, although a full understanding of intrinsic factors is only partially supported. The work provides a **convincing** and **important** molecular dataset, and strengthens our understanding of intrinsic regulation, and would be of interest more broadly to cell biologists seeking to understand immune cell development.

---

## [Referee Report · Reviewer #1 (Public Review)]

The study starts with the notion that in an AD-like disease model, ILC2s in the Rag1 knock-out were expanded and contained relatively more IL-5+ and IL-13+ ILC2s. This was confirmed in the Rag2 knock-out mouse model.

By using a chimeric mouse model in which wild-type knock-out splenocytes were injected into irradiated Rag1 knock-out mice, it was shown that even though the adaptive lymphocyte compartment was restored, there were increased AD-like symptoms and increased ILC2 expansion and activity. Moreover, in the reverse chimeric model, i.e. injecting a mix of wild-type and Rag1 knock-out splenocytes into irradiated wild-type animals, it was shown that the Rag1 knock-out ILC2s expanded more and were more active. Therefore, the authors could conclude that the RAG1 mediated effects were ILC2 cell-intrinsic.

Subsequent fate-mapping experiments using the Rag1Cre;reporter mouse model showed that there were indeed RAGnaïve and RAGexp ILC2 populations within naïve mice. Lastly, the authors performed multi-omic profiling, using single-cell RNA sequencing and ATAC-sequencing, in which a specific gene expression profile was associated with ILC2. These included well-known genes but the authors notably also found expression of Ccl1 and Ccr8 within the ILC2. The authors confirmed their earlier observations that in the RAGexp ILC2 population, the Th2 regulome was more suppressed, i.e. more closed, compared to the RAGnaïve population, indicative of the suppressive function of RAG on ILC2 activity. I do agree with the authors' notion that the main weakness was that this study lacks the mechanism by which RAG regulates these changes in ILC2s.

The manuscript is very well written and easy to follow, and the compelling conclusions are well supported by the data. The experiments are meticulously designed and presented. I wish to commend the authors for the study's quality.

---

## [Referee Report · Reviewer #2 (Public Review)]

Summary:

The study by Ver Heul et al., investigates the consequences of RAG expression for type 2 innate lymphoid cell (ILC2) function. RAG expression is essential for the generation of the receptors expressed by B and T cells and their subsequent development. Innate lymphocytes, which arise from the same initial progenitor populations, are in part defined by their ability to develop in the absence of RAG expression. However, it has been described in multiple studies that a significant proportion of innate lymphocytes show a history of Rag expression. In compelling studies several years ago, members of this research team revealed that early Rag expression during the development of Natural Killer cells (Karo et al., Cell 2014), the first described innate lymphocyte, had functional consequences.

Here, the authors revisit this topic, a worthwhile endeavour given the broad history of Rag expression within all ILCs and the common use of RAG-deficient mice to specifically assess ILC function. Focusing on ILC2s and utilising state-of-the-art approaches, the authors sought to understand whether early expression of Rag during ILC2 development had consequences for activity, fitness, or function. Having identified cell-intrinsic effects in vivo, the authors investigated the causes of this, identifying epigenetic changes associated with the accessibility genes associated with core ILC2 functions.

The manuscript is well written and does an excellent job of supporting the reader through reasonably complex transcriptional and epigenetic analyses, with considerate use of explanatory diagrams. Overall I think that the conclusions are fair, the topic is thought-provoking, and the research is likely of broad immunological interest. I think that the extent of functional data and mechanistic insight is appropriate.

Strengths:

- The logical and stepwise use of mouse models to first demonstrate the impact on ILC2 function in vivo and a cell-intrinsic role. Initial analyses show enhanced cytokine production by ILC2 from RAG-deficient mice. Then through two different chimeric mice (including BM chimeras), the authors convincingly show this is cell intrinsic and not simply as a result of lymphopenia. This is important given other studies implicating enhanced ILC function in RAG-/- mice reflect altered competition for resources (e.g. cytokines).

- Use of Rag expression fate mapping to support analyses of how cells were impacted - this enables a robust platform supporting subsequent analyses of the consequences of Rag expression for ILC2.

- Use of snRNA-seq supports gene expression and chromatin accessibility studies - these reveal clear differences in the data sets consistent with altered ILC2 function.

- Convincing evidence of epigenetic changes associated with loci strongly linked to ILC2 function. This forms a detailed analysis that potentially helps explain some of the altered ILC2 functions observed in ex vivo stimulation assays.

- Provision of a wealth of expression data and bioinformatics analyses that can serve as valuable resources to the field.

---

## [Author Response]

The following is the authors’ response to the original reviews

**Public Reviews:**

**Reviewer #1 (Public Review):**
The study starts with the notion that in an AD-like disease model, ILC2s in the Rag1 knockout were expanded and contained relatively more IL-5^+^ and IL-13^+^ ILC2s. This was confirmed in the Rag2 knock-out mouse model.By using a chimeric mouse model in which wild-type knock-out splenocytes were injected into irradiated Rag1 knock-out mice, it was shown that even though the adaptive lymphocyte compartment was restored, there were increased AD-like symptoms and increased ILC2 expansion and activity. Moreover, in the reverse chimeric model, i.e. injecting a mix of wild-type and Rag1 knock-out splenocytes into irradiated wild-type animals, it was shown that the Rag1 knock-out ILC2s expanded more and were more active. Therefore, the authors could conclude that the RAG1 mediated effects were ILC2 cell-intrinsic.Subsequent fate-mapping experiments using the Rag1Cre;reporter mouse model showed that there were indeed RAGnaïve and RAGexp ILC2 populations within naïve mice. Lastly, the authors performed multi-omic profiling, using single-cell RNA sequencing and ATACsequencing, in which a specific gene expression profile was associated with ILC2. These included well-known genes but the authors notably also found expression of Ccl1 and Ccr8 within the ILC2. The authors confirmed their earlier observations that in the RAGexp ILC2 population, the Th2 regulome was more suppressed, i.e. more closed, compared to the RAGnaïve population, indicative of the suppressive function of RAG on ILC2 activity. I do agree with the authors' notion that the main weakness was that this study lacks the mechanism by which RAG regulates these changes in ILC2s.The manuscript is very well written and easy to follow, and the compelling conclusions are well supported by the data. The experiments are meticulously designed and presented. I wish to commend the authors for the study's quality.Even though the study is compelling and well supported by the presented data, some additional context could increase the significance:(1) The presence of the RAGnaïve and RAGexp ILC2 populations raises some questions on the (different?) origin of these populations. It is known that there are different waves of ILC2 origin (most notably shown in the Schneider et al Immunity 2019 publication, PMID 31128962). I believe it would be very interesting to further discuss or possibly show if there are different origins for these two ILC populations.Several publications describe the presence and origin of ILC2s in/from the thymus (PMIDs 33432227 24155745). Could the authors discuss whether there might be a common origin for the RAGexp ILC2 and Th2 cells from a thymic lineage? If true that the two populations would be derived from different populations, e.g. being the embryonic (possibly RAGnaïve) vs. adult bone marrow/thymus (possibly RAGexp), this would show a unique functional difference between the embryonic derived ILC2 vs. adult ILC2.

We agree with the Reviewer that our findings raise important questions about ILC ontogeny. These are areas of ongoing investigation for us, and it is our hope this study may inform further investigation by others as well.

Regarding the Schneider et al study, we have considered the possibility that RAG expression may mark a particular wave of ILC2 origin. In that study, the authors used a tamoxifen-based inducible Cre strategy in their experiments to precisely time the lineage tracing of a reporter from the Rosa26 locus. Those lineage tracing mice would overlap genetically with the RAG lineage tracing mice we used in our current study, thus performing combined timed migration fate mapping and RAG fate mapping experiments would require creating novel mouse strains.

Similarly, the possible influence of the thymic or bone marrow environment on RAG expression in ILCs is an exciting possibility. Perhaps there are signals common to those environments that can influence all developing lymphocytes, including not only T and B cells but also ILCs, with one consequence being induction of RAG expression. While assessing levels of RAG-experienced ILCs in these tissues using our lineage tracing mouse may hint at these possibilities, conclusive evidence would require more precise control over the timing of RAG lineage tracing than our current reagents allow (e.g. to control for induction in those environments vs migration of previously fate-mapped cells to those environments).

To answer these questions directly, we are developing orthogonal lineage tracing mouse strains, which can report on both timing of ILC development and RAG expression, but these mice are not available yet. Given the limitations of our currently available reagents, we were careful to focus our manuscript on the skin phenotype and the more descriptive aspects of the RAG-induced phenotype. We have elaborated on these important questions and referenced all the studies noted by the Reviewer in the Discussion section as areas of future inquiry on lines 421-433.

(2) On line 104 & Figures 1C/G etc. the authors describe that in the RAG knock-out ILC2 are relatively more abundant in the lineage negative fraction. On line 108 they further briefly mentioned that this observation is an indication of enhanced ILC2 expansion. Since the study includes an extensive multi-omics analysis, could the authors discuss whether they have seen a correlation of RAG expression in ILC2 with regulation of genes associated with proliferation, which could explain this phenomenon?

We thank the Reviewer for pointing out this opportunity to further correlate our functional and multiomic findings. To address this, we first looked deeper into our prior analyses and found that among the pathways enriched in GSEA analysis of differentially expressed genes (DEGs) between RAG^+^ and RAG^-^ ILC2s, one of the pathways suppressed in RAG^+^ ILC2s was “GOBP_EPITHELIAL_CELL_PROLIFERATION.”

(Author response image 1). There are a few other gene sets present in other databases such as MSigDB with terms including “proliferation,” but these are often highly specific to a particular cell type and experimental or disease condition (e.g. tissue-specific cancers). We did not find any of these enriched in our GSEA analysis.

**Author response image 1. sa3fig1:** GSEA plot of GOBP epithelial proliferation pathway in RAG-experienced vs RAG-naïve ILC2s.

The ability to predict cellular proliferation states from transcriptomic data is an area of active research, and there does not appear to be any universally accepted method to do this reliably. We found two recent studies (PMIDs 34762642; 36201535) that identified novel “proliferation signatures.” Since these gene sets are not present in any curated database, we repeated our GSEA analysis using a customized database with the addition of these gene sets. However, we did not find enrichment of these sets in our RAG+/- ILC2 DEG list. We also applied our GPL strategy integrating analysis of our epigenomic data to the proliferation signature genes, but we did not see any clear trend. Conversely, our GSEA analysis did not identify any enrichment for apoptotic signatures as a potential mechanism by which RAG may suppress ILC2s.

Notwithstanding the limitations of inferring ILC2 proliferation states from transcriptomic and epigenomic data, our experimental data suggest RAG exerts a suppressive effect on ILC2 proliferation. To formally test the hypothesis that RAG suppresses proliferation in the most rigorous way, we feel new mouse strains are needed that allow simultaneous RAG fate mapping and temporally restricted fate mapping. We elaborate on this in new additions to the discussion on lines 421-433.

**Reviewer #2 (Public Review):**
Summary:The study by Ver Heul et al., investigates the consequences of RAG expression for type 2 innate lymphoid cell (ILC2) function. RAG expression is essential for the generation of the receptors expressed by B and T cells and their subsequent development. Innate lymphocytes, which arise from the same initial progenitor populations, are in part defined by their ability to develop in the absence of RAG expression. However, it has been described in multiple studies that a significant proportion of innate lymphocytes show a history of Rag expression. In compelling studies several years ago, members of this research team revealed that early Rag expression during the development of Natural Killer cells (Karo et al., Cell 2014), the first described innate lymphocyte, had functional consequences.Here, the authors revisit this topic, a worthwhile endeavour given the broad history of Rag expression within all ILCs and the common use of RAG-deficient mice to specifically assess ILC function. Focusing on ILC2s and utilising state-of-the-art approaches, the authors sought to understand whether early expression of Rag during ILC2 development had consequences for activity, fitness, or function. Having identified cell-intrinsic effects in vivo, the authors investigated the causes of this, identifying epigenetic changes associated with the accessibility genes associated with core ILC2 functions.The manuscript is well written and does an excellent job of supporting the reader through reasonably complex transcriptional and epigenetic analyses, with considerate use of explanatory diagrams. Overall I think that the conclusions are fair, the topic is thoughtprovoking, and the research is likely of broad immunological interest. I think that the extent of functional data and mechanistic insight is appropriate.Strengths:- The logical and stepwise use of mouse models to first demonstrate the impact on ILC2 function in vivo and a cell-intrinsic role. Initial analyses show enhanced cytokine production by ILC2 from RAG-deficient mice. Then through two different chimeric mice (including BM chimeras), the authors convincingly show this is cell intrinsic and not simply as a result of lymphopenia. This is important given other studies implicating enhanced ILC function in RAG-/- mice reflect altered competition for resources (e.g. cytokines).- Use of Rag expression fate mapping to support analyses of how cells were impacted - this enables a robust platform supporting subsequent analyses of the consequences of Rag expression for ILC2.- Use of snRNA-seq supports gene expression and chromatin accessibility studies - these reveal clear differences in the data sets consistent with altered ILC2 function.- Convincing evidence of epigenetic changes associated with loci strongly linked to ILC2 function. This forms a detailed analysis that potentially helps explain some of the altered ILC2 functions observed in ex vivo stimulation assays.- Provision of a wealth of expression data and bioinformatics analyses that can serve as valuable resources to the field.

We appreciate the strengths noted by the Reviewer for our study. We would like to especially highlight the last point about our single cell dataset and provision of supplemental data tables. Although our study is focused on AD-like skin disease and skin draining lymph nodes, we hope that our findings can serve as a valuable resource for future investigation into mechanisms of RAG modulation of ILC2s in other tissues and disease states.

Weaknesses:- Lack of insight into precisely how early RAG expression mediates its effects, although I think this is beyond the scale of this current manuscript. Really this is the fundamental next question from the data provided here.

We thank the Reviewer for their recognition of the context of our current work and its future implications. We aimed to present compelling new observations within the scope of what our current data can substantiate. We believe answering the next fundamental question of the mechanisms by which RAG mediates its effects in ILC2s will require development of novel reagents. We are actively pursuing this, and we look forward to others building on our findings as well.

- The epigenetic analyses provide evidence of differences in the state of chromatin, but there is no data on what may be interacting or binding at these sites, impeding understanding of what this means mechanistically.

We thank the Reviewer for pointing out this aspect of the epigenomic data analysis and the opportunity to expand the scope of our manuscript. We performed additional analyses of our data to identify DNA binding motifs and infer potential transcription factors that may be driving the effects of a history of RAG expression that we observed. We hope that these additional data, analyses, and interpretation add meaningful insight for our readers.

We first performed the analysis for the entire dataset and validated that the analysis yielded results consistent with prior studies (e.g. finding EOMES binding motifs as a marker in NK cells). Then, we examined the differences in RAG fate-mapped ILC2s. These analyses are in new Figure S10 and discussed on lines 277-316.

We also performed an analysis specifically on the Th2 locus, given the effects of RAG on type 2 cytokine expression. These analyses are in new Figure S12 and discussed on lines 366-378.

- Focus on ILC2 from skin-draining lymph nodes rather than the principal site of ILC2 activity itself (the skin). This may well reflect the ease at which cells can be isolated from different tissues.

We appreciate the Reviewer’s insight into the limitations of our study. Difficulties in isolating ILC2s from the skin were indeed a constraint in our study. In particular, we were unable to isolate enough ILC2s from the skin for stimulation and cytokine staining. Given that one of our main hypotheses was that RAG affects ILC2 function, we focused our studies on skin draining lymph nodes, which allowed measurement of the two main ILC2 functional cytokines, IL-5 and IL-13, as readouts in the key steady state and AD-like disease experiments.

- Comparison with ILC2 from other sites would have helped to substantiate findings and compensate for the reliance on data on ILC2 from skin-draining lymph nodes, which are not usually assessed amongst ILC2 populations.

We agree with the Reviewer that a broader survey of the RAG-mediated phenotype in other tissues and by extension other disease models would strengthen the generalizability of our observations. Indeed, we did a more expansive survey of tissues in our BM chimera experiments. We found a similar trend to our reported findings in the sdLN in tissues known to be affected by ILC2s (Author response image 2) including the skin and lung and in other lymphoid tissues including spleen and mesenteric lymph nodes (mLN). We found that donor reconstitution in each tissue was robust except for the skin, where there was no significant difference between host and -donor CD45^+^ immune cells and where CD45^-^ parenchymal cells predominated (Author response image 2A,C,E,G,I). This may explain why *Rag1-/-* donor ILC2s were significantly higher in proportion in all tissues except the skin, where we observed a similar trend that was not statistically significant (Author response image 2B,D,F,H,J).

Notwithstanding these results, given that we unexpectedly observed enhanced AD-like inflammation in the MC903 model in *Rag1* KO mice, we concentrated our later experiments and analyses on defining the differences in skin draining ILC2s modulated by RAG. Our subsequent findings in the skin provoke many new hypotheses about the role of RAG in ILC2s in other tissues, and our tissue survey in the BM chimera provides additional rationale to pursue similar studies in disease models in other tissues. While this is an emerging area of investigation in our lab, we opted to focus this manuscript on our findings related to the AD-like disease model. We have ongoing studies to investigate other tissues, and we are still in the early stages of developing disease models to expand on these findings. However, if the reviewer feels strongly this additional data should be included in the manuscript, we are happy to add it. Considering the complexity of the data and concepts in the manuscript, we hoped to keep it focused to where we have strong molecular, cellular, and phenotypic outcomes.

**Author response image 2. sa3fig2:** Comparison of immune reconstitution in and ILC2 donor proportions in different tissues from BM chimeras. Equal quantities of bone marrow cells from *Rag1-/-* (CD45.2,CD90.2) and WT (CD45.2, CD90.1) C57BL/6J donor mice were used to reconstitute the immune systems of irradiated recipient WT (CD45.1) C57BL/6J mice. The proportion of live cells that are donor-derived (CD45.2), host-derived (CD45.1), or parenchymal (CD45-) [above] and proportion of ILC2s that are from *Rag1-/-* (CD90.2) or WT (CD90.1) donors [below] for (A,B) skin (C,D) sdLN (E,F) lung (G,H) spleen and (I,J) mLN.

- The studies of how ILC2 are impacted are a little limited, focused exclusively on IL-13 and IL-5 cytokine expression.

We agree with the reviewer that our functional readout on IL-5 and IL-13 is relatively narrow. However, this focused experimental design was based on several considerations. First, IL-5 and IL-13 are widely recognized as major ILC2 effector molecules (Vivier et al, 2018, PMID 30142344). Second, in the MC903 model of AD-like disease, we have previously shown a clear correlation between ILC2s, levels of IL-5 and IL-13, and disease severity as measured by ear thickness (Kim et al, 2013, PMID 23363980). Depletion of ILC2s led to decreased levels of IL-13 and IL-5 and correspondingly reduced ear inflammation. However, while ILC2s are also recognized to produce other effector molecules such as IL-9 and Amphiregulin, which are likely involved in human atopic dermatitis (Namkung et al, 2011, PMID 21371865; Rojahn et al, 2020, PMID 32344053), there is currently no evidence linking these effectors to disease severity in the MC903 model. Third, IL-13 is emerging as a key cytokine driving atopic dermatitis in humans (Tsoi et al, 2019, PMID 30641038). Drugs targeting the IL-4/IL-13 receptor (dupilumab), or IL-13 itself (tralokinumab, lebrikizumab), have shown clear efficacy in treating atopic dermatitis. Interestingly, drugs targeting more upstream molecules, like TSLP (tezepelumab) or IL-33 (etokimab), have failed in atopic dermatitis. Taken together, these findings from both mouse and human studies suggest IL-13 is a critical therapeutic target, and thus functional readout, in determining the clinical implications of type 2 immune activation in atopic dermatitis.

Aside from effector molecules, other readouts such as surface receptors may be of interest in understanding the mechanism of how RAG influences ILC2 function. For example, IL-18 has been shown to be an important co-stimulatory molecule along with TSLP in driving production of IL-13 by cutaneous ILC2s (Ricardo-Gonzalez et al, 2018, PMID 30201992). Our multiomic analysis showed decreased IL-18 receptor regulome activity in RAG-experienced ILC2s, which may be a mechanism by which RAG suppresses IL-13 production. Ultimately, in that study the role of IL-18 in enhancing MC903-induced inflammation through ILC2s was via increased production of IL-13, which was one of our major functional readouts. To clearly define mechanisms like these will require generation of new mice to interrogate RAG status in the context of tissue-specific knockout of other genes, such as the IL-18 receptor. We plan to perform these types of experiments in follow up studies. Notwithstanding this, we have now included additional discussion on lines 476508 to highlight why understanding how RAG impacts other regulatory and effector pathways would be an interesting area of future inquiry.

**Reviewer #3 (Public Review):**
In this study, Ver Heul et al. investigate the role of RAG expression in ILC2 functions. While RAG genes are not required for the development of ILCs, previous studies have reported a history of expression in these cells. The authors aim to determine the potential consequences of this expression in mature cells. They demonstrate that ILC2s from RAG1 or RAG2 deficient mice exhibit increased expression of IL-5 and IL-13 and suggest that these cells are expanded in the absence of RAG expression. However, it is unclear whether this effect is due to a direct impact of RAG genes or a consequence of the lack of T and B cells in this condition. This ambiguity represents a key issue with this study: distinguishing the direct effects of RAG genes from the indirect consequences of a lymphopenic environment.The authors focus their study on ILC2s found in the skin-draining lymph nodes, omitting analysis of tissues where ILC2s are more enriched, such as the gut, lungs, and fat tissue. This approach is surprising given the goal of evaluating the role of RAG genes in ILC2s across different tissues. The study shows that ILC2s derived from RAG-/- mice are more activated than those from WT mice, and RAG-deficient mice show increased inflammation in an atopic dermatitis (AD)-like disease model. The authors use an elegant model to distinguish ILC2s with a history of RAG expression from those that never expressed RAG genes. However, this model is currently limited to transcriptional and epigenomic analyses, which suggest that RAG genes suppress the type 2 regulome at the Th2 locus in ILC2s.

We agree with the Reviewer that understanding the role of RAG in ILC2s across different tissues is an important goal. One of the primary inspirations for our paper was the clinical paradox that patients with Omenn syndrome, despite having profound adaptive T cell deficiency, develop AD with much greater penetrance than in the general population. Thus, there was always an appreciation for the likelihood that skin ILC2s have a unique proclivity towards the development of AD-like disease. Notwithstanding this, given the profound differences that can be found in ILC2s based on their tissue residence and disease state (as the Reviewer also points out below), we focused our investigations on characterizing the skin draining lymph nodes to better define factors underlying our initial observations of enhanced AD-like disease in *Rag1-/-* mice. While our findings in skin provoke the hypothesis that similar effects may be observed in other tissues and influence corresponding disease states, we were cautious not to suggest this may be the case by reporting surveys of other tissues without development of additional disease models to formally test these hypotheses. We present this manuscript now as a short, skin-focused study, rather than delaying publication to expand its scope. Truthfully, this project started in 2015 and has undergone many delays with the hopes of newer technologies and reagents coming to add greater clarity. We hope our study will enable others to pursue the goal of understanding the broader effects of RAG in ILC2s, and potentially other innate lymphoid lineages as well.

We did a more expansive survey of tissues in our BM chimera experiments. We found a similar trend to our reported findings in the sdLN in tissues known to be affected by ILC2s (Author response image 2) including the skin and lung and in other lymphoid tissues including spleen and mesenteric lymph nodes (mLN). We found that donor reconstitution in each tissue was robust except for the skin, where there was no significant difference between host and donor CD45^+^ immune cells and where CD45^-^ parenchymal cells predominated (Author response image 2A,C,E,G,I). This may explain why *Rag1-/-* donor ILC2s were significantly higher in proportion in all tissues except the skin, where we observed a similar trend that was not statistically significant (Author response image 2B,D,F,H,J). However, given the lack of correlation to disease readouts in other organ systems, we chose to not include this data in our manuscript. However, if the Reviewer feels these data should be included, we would be happy to include as a supplemental figure.

The authors report a higher frequency of ILC2s in RAG-/- mice in skin-draining lymph nodes, which is expected as these mice lack T and B cells, leading to ILC expansion. Previous studies have reported hyper-activation of ILCs in RAG-deficient mice, suggesting that this is not necessarily an intrinsic phenomenon. For example, RAG-/- mice exhibit hyperphosphorylation of STAT3 in the gut, leading to hyperactivation of ILC3s. This study does not currently provide conclusive evidence of an intrinsic role of RAG genes in the hyperactivation of ILC2s. The splenocyte chimera model is artificial and does not reflect a normal environment in tissues other than the spleen. Similarly, the mixed BM model does not demonstrate an intrinsic role of RAG genes, as RAG1-/- BM cells cannot contribute to the B and T cell pool, leading to an expected expansion of ILC2s. As the data are currently presented it is expected that a proportion of IL-5-producing cells will come from the RAG1/- BM.

The Reviewer raises an important point about the potential cell-intrinsic roles of RAG vs the many cell-extrinsic explanations that could affect ILC2 populations, with the most striking being the lack of T and B cells in RAG knockout mice. It is well-established that splenocyte transfer into T and B cell-deficient mice reconstitutes T cell-mediated effects (such as the T cell transfer colitis model pioneered by Powrie and others), and we were careful in our interpretation of the splenocyte chimera experiment to conclude only that lack of Tregs was unlikely to explain the enhanced ADlike disease in T (and B) cell-deficient mice.

We agree with the Reviewer that the *Rag1-/-* BM will not contribute to the B and T cell pool. However, BM from the WT mice would be expected to contribute to development of the adaptive lymphocyte pool. Indeed, we found that most of the CD45^+^ immune cells in the spleens of BM chimera mice were donor-derived (Author response image 3A), and total levels of B cells and T cells showed reconstitution in a pattern similar to control spleens from donor WT mice, while spleens from donor *Rag1-/-* mice expectedly had essentially no detectable adaptive lymphocytes (Author response image 3B-D). From this, we concluded the BM chimera experiment was successful in establishing an immune environment with the presence of adaptive lymphocytes, and the differences in ILC2 proportions we observed were in the context of developing alongside a normal number of B and T lymphocytes. Notwithstanding the potential role of the adaptive lymphocyte compartment in shaping ILC2 development, since we transplanted equal amounts of WT and *Rag1-/-* BM into the same recipient environment, we are not able to explain how cell-extrinsic effects alone would account for the unequal numbers of WT vs *Rag1-/-* ILC2s we observed after immune reconstitution.

**Author response image 3. sa3fig3:** Comparison of immune reconstitution in BM chimeras to controls. Equal quantities of bone marrow cells from *Rag1-/-* (CD45.2) and WT (CD45.2) C57BL/6J donor mice were used to reconstitute the immune systems of irradiated recipient WT (CD45.1) C57BL/6J mice. (A) Number of WT recipient CD45.1+ immune cells in the spleens of recipient mice compared to number of donor CD45.2+ cells (WT and *Rag1-/-*) normalized to 100,000 live cells. Comparison of numbers of B cells, CD4+ T cells, and CD8+ T cells in spleens of (B) BM chimera mice, (C) control WT mice and (D) control *Rag1-/-* mice.

We also subsequently found transcriptional and epigenomic differences in RAG-experienced ILC2s compared to RAG-naïve ILC2s. Critically, these differences were present in ILC2s from the same mice that had developed normally within an intact immune system, rather than in the setting of a BM transplant or a defective immune background such as in *Rag1-/-* mice.

We recognize that there are almost certainly cell-extrinsic factors affecting ILC2s in *Rag1-/-* mice due to lack of B and T cells, and that BM chimeras are not perfect substitutes for simulating normal hematopoietic development. However, the presence of cell-extrinsic effects does not negate the potential contribution of cell-intrinsic factors as well, and we respectfully stand by our conclusion that our data support a role, however significant, for cell-intrinsic effects of RAG in ILC2s.

Finally, the Reviewer mentions the interesting observation that gut ILC3s exhibit hyperphosphorylation of STAT3 in *Rag1-/-* mice compared to WT as an example of cell-extrinsic effects of RAG deficiency (we assume this is in reference to Mao et al, 2018, PMID 29364878 and subsequent work). We now reference this paper and have included additional discussion on how our observations of ILC2s may be generalizable to not only other organ systems, but also other ILC subsets, limitations on these generalizations, and future directions on lines 477-520.

Overall, the level of analysis could be improved. Total cell numbers are not presented, the response of other immune cells to IL-5 and IL-13 (except the eosinophils in the splenocyte chimera mice) is not analyzed, and the analysis is limited to skin-draining lymph nodes.

We thank the Reviewer for the suggestions to add rigor to our analysis. ILC2 populations are relatively rare, and we designed our experiments to assess frequencies, rather than absolute numbers. We did not utilize counting beads, so our counts may not be comparable between samples. We have added additional data for absolute cell counts normalized to 100,000 live cells for each experiment (see below for a summary of new panels in each figure). Our new data on total cell numbers are consistent with the initial observations regarding frequency of ILC2s we reported from our experiments. For the BM chimera experiments, we presented the proportions of ILC2s, and IL-5 and IL-13 positive ILC2s, by donor source, as this is the critical question of the experiment. Notwithstanding our analysis by proportion, we found that the frequency of *Rag1-/-* ILC2s, IL-5^+^ cells, or IL-13^+^ cells within Lin- population was also significantly increased. While our initial submission included only the proportions for clarity and simplicity, we now include frequency and absolute numbers in new panels for more critical appraisal of our data by readers.

In New Figure 1, we added new panels for ILC2 cell number in both the AD-like disease experiment (C) and in steady state (H).

In New Figure S2, we added a panel for ILC2 cell number in steady state (B).

In Figure 2 and associated supplemental data in Figure S4, we added several more panels. For the splenocyte chimera, we added a panel for ILC2 cell number in New Figure 2C.

We incorporated multiple new panels in New Figure S4 to address the need for more data to be shown for the BM chimera (also requested by Reviewer #2). These included total cell counts and frequency for ILC2 (New Figure S4F,G), and IL-5^+^ (New Figure S4I,K) and IL-13^+^ (New Figure S4J,L) ILCs in addition to the proportions originally presented in Figure 2.

In terms of the limited analysis of other tissues, our initial observation of enhanced AD-like disease in *Rag1-/-* compared to WT mice built on our prior work elucidating the role of ILC2s in the MC903 model of AD-like disease in mice and AD in humans (Kim et al, 2013, PMID 23363980). Consequently, we focused on the skin to further develop our understanding of the role of RAG1 in this model. As in our prior studies, technical limitations in obtaining sufficient numbers of ILC2s from the skin itself for ex vivo stimulation to assess effector cytokine levels required performing these experiments in the skin draining lymph nodes.

We agree that IL-5 and IL-13 are major mediators of type 2 pathology and studying their effects on immune cells is an important area of inquiry, particularly since there are multiple drugs available or in development targeting these pathways. However, our goal was not to study what was happening downstream of increased cytokine production from ILC2s, but instead to understand what was different about RAG-deficient or RAG-naïve ILC2s themselves that drive their expansion and production of effector cytokines compared to RAG-sufficient or RAGexperienced ILC2s. By utilizing the same MC903 model in which we previously showed a critical role for ILC2s in driving IL-5 and IL-13 production and subsequent inflammation in the skin, we were able to instead focus on defining the cell-intrinsic aspects of RAG function in ILC2s.

The authors have a promising model in which they can track ILC2s that have expressed RAG or not. They need to perform a comprehensive characterization of ILC2s in these mice, which develop in a normal environment with T and B cells. Approximately 50% of the ILC2s have a history of RAG expression. It would be valuable to know whether these cells differ from ILC2s that never expressed RAG, in terms of proliferation and expression of IL5 and IL-13. These analyses should be conducted in different tissues, as ILC2s adapt their phenotype and transcriptional landscape to their environment. Additionally, the authors should perform their AD-like disease model in these mice.

We agree with the Reviewer (and a similar comment from Reviewer #2) that a broader survey of the RAG-mediated phenotype in other tissues and by extension other disease models would strengthen the generalizability of our observations. Indeed, we did a more expansive survey of tissues in our BM chimera experiments. We found a similar trend to our reported findings in the sdLN in tissues known to be affected by ILC2s (Author response image 2) including the skin and lung and in other lymphoid tissues including spleen and mesenteric lymph nodes (mLN). We found that donor reconstitution in each tissue was robust except for the skin, where there was no significant difference between host and donor CD45^+^ immune cells and where CD45^-^ parenchymal cells predominated (Author response image 2A,C,E,G,I). This may explain why *Rag1-/-* donor ILC2s were significantly higher in proportion in all tissues except the skin, where we observed a similar trend that was not statistically significant (Author response image 2B,D,F,H,J). We omitted these analyses to maintain the focus on the skin, but we will be happy to add this data to the manuscript if the Reviewer feels this figure should be helpful.

Notwithstanding these results, given that we unexpectedly observed enhanced AD-like inflammation in the MC903 model in *Rag1* KO mice, we concentrated our later experiments and analyses on defining the differences in skin draining ILC2s modulated by RAG. Our subsequent findings in the skin provoke many new hypotheses about the role of RAG in ILC2s in other tissues, and our tissue survey in the BM chimera provides additional rationale to pursue similar studies in disease models in other tissues. While this is an emerging area of investigation in our lab, we opted to focus this manuscript on our findings related to the AD-like disease model. We have ongoing studies to investigate other tissues, and we are still in the early stages of developing disease models to expand on these findings. However, if the reviewer feels strongly this additional data should be included in the manuscript, we are happy to add it. Considering the complexity of the data and concepts in the manuscript, we hoped to keep it focused to where we have strong molecular, cellular, and phenotypic outcomes. We elaborate on the implications of our work for future studies, including limitations of our study and currently available reagents and need for new mouse strains to rigorously answer these questions on lines 476-508

The authors provide a valuable dataset of single-nuclei RNA sequencing (snRNA-seq) and ATAC sequencing (snATAC-seq) from RAGexp (RAG fate map-positive) and RAGnaïve (RAG fate map-negative) ILC2s. This elegant approach demonstrates that ILC2s with a history of RAG expression are epigenomically suppressed. However, key genes such as IL-5 and IL-13 do not appear to be differentially regulated between RAGexp and RAGnaïve ILC2s according to Table S5. Although the authors show that the regulome activity of IL-5 and IL-13 is decreased in RAGexp ILC2s, how do the authors explain that these genes are not differentially expressed between the RAGexp and RAGnaïve ILC2? I think that it is important to validate this in vivo.

We thank the Reviewer for highlighting the value and possible elegance of our data. The Reviewer brings up an important issue that we grappled with in this study and that highlights a major technical limitation of single cell sequencing studies. Genes for secreted factors such as cytokines are often transcribed at low levels and are poorly detected in transcriptomic studies. This is particularly true in single cell studies with lower sequencing depth. Various efforts have been made to overcome these issues such as computational approaches to estimate missing data (e.g. van Djik et al, 2018, PMID 29961576; Huang et al, 2018, PMID 29941873), or recent use of cytokine reporter mice and dial-out PCR to enhance key cytokine signals in sequenced ILCs (Bielecki et al, 2021, PMID 33536623). We did not utilize computational methods to avoid the risk of introducing artifacts into the data, and we did not perform our study in cytokine reporter mice. Thus, cytokines were poorly detected in our transcriptomic data, as evidenced by lack of identification of cytokines as markers for specific clusters (e.g. IL-5 for ILC2s) or significant differential expression between RAG-naïve and RAG-experienced ILC2s.

However, the multiomic features of our data allowed a synergistic analysis to identify effects on cytokines. For example, transcripts for the IL-4 and IL-5 were not detected at a high enough level to qualify as marker genes of the ILC2 cluster in the gene expression (GEX) assay but were identified as markers for the ILC2 cluster in the ATAC-seq data in the differentially accessible chromatin (DA) assay. Using the combined RNA-seq and ATAC-seq gene to peak links (GPL) analyses, many GPLs were identified in the Th2 locus for ILC2s, including for IL-13, which was not identified as a marker for ILC2s by any of the assays alone. Thus, our combined analysis took advantage of the potential of multiomic datasets to overcome a general weakness inherent to most scRNAseq datasets.

**Recommendations for the authors:**

**Reviewer #1 (Recommendations For The Authors):**
- Line 168; Reference 23 also showed expression in the NK cells, please add this reference to reference 24.

We thank the reviewer for catching this oversight, and we have corrected it in the revised manuscript.

- Please add the full names for GPL and sdLN in the text of the manuscript when first using these abbreviations. They are now only explained in the legends.

We reviewed the manuscript text and found that we defined sdLNs for the first time on line 104. We defined GPLs for the first time on line 248. We believe these definitions are placed appropriately near the first references to the corresponding figures/analysis, but if the Reviewer believes we should move these definitions earlier, we are happy to do so.

**Reviewer #2 (Recommendations For The Authors):**
I would suggest that the following reanalyses would improve the clarity of the data:- Can ILC2 numbers, rather than frequency, be used (e.g. in Figure 1C, S2B, and so on). This would substantiate the data that currently relies on percentages.

This was a weakness also noted by Reviewer #3. We have added data on ILC2 numbers for each experiment as outlined below:

In New Figure 1, we added new panels for ILC2 cell number in both the AD-like disease experiment (C) and in steady state (H).

In New Figure S2, we added a panel for ILC2 cell number in steady state (B).

In Figure 2 and associated supplemental data in Figure S4, we added several more panels. For the splenocyte chimera, we added a panel for ILC2 cell number in New Figure 2C.

We incorporated multiple new panels in New Figure S4 to address the need for more data to be shown for the BM chimera (also requested by Reviewer #2). These included total cell counts and frequency for ILC2 (New Figure S4F,G), and IL-5^+^ (New Figure S4I,K) and IL-13^+^ (New Figure S4J,L) ILCs in addition to the proportions originally presented in Figure 2.

- Can the authors provide data on IL-33R expression on sdLN ILC2s? Expression of ST-2 (IL-33R) does vary between ILC2 populations and is impacted by the digestion of tissue. All of the data provided here requires ILC2 to be IL-33R^+^. In the control samples, the ILC2 compartment is very scarce - in LNs, ILC2s are rare. The gating strategy with limited resolution of positive and negative cells in the lineage gate doesn't help this analysis.

The Reviewer raises a valid point regarding the IL-33R marker and ILC2s. We designed our initial experiments to be consistent with our earlier observations of skin ILC2s, which were defined as CD45^+^Lin-CD90+CD25+IL33+, and the scarcity of skin draining lymph node ILC2s at steady state was consistent with our prior findings (Kim et al, 2013, PMID 23363980). We can include MFI data on IL-33R expression in these cells if the reviewer feels strongly that this would add to the manuscript, but we did not include other ILC2-specific markers in these experiments that would give us an alternative total ILC2 count to calculate frequency of IL-33R^+^ ILC2s, which would also make the context of the IL-33 MFI difficult to interpret.

Other studies defining tissue specific expression patterns in ILC2s have called into question whether IL-33R is a reliable marker to define skin ILC2s (Ricardo-Gonzalez et al, 2018, PMID 30201992). However, there is evidence for region-specific expression of IL-33R (Kobayashi et al, 2019, PMID 30712873), with ILC2s in the subcutis expressing high levels of IL-33R and both IL5 and IL-13, while ILC2s in the epidermis and dermis have low levels of IL-33R and IL-5 expression. In contrast to the Kobayashi et al study, Ricardo-Gonzalez et al sequenced ILC2s from whole skin, thus the region-specific expression patterns were not preserved, and the lower expression of IL-33R in the epidermis and dermis may have diluted the signal from the ILC2s in the subcutis. These may also be the ILC2s most likely to drain into the lymph nodes, which is the tissue on which we focused our analyses (consistent with our prior work in Kim et al, 2013).

- In Figure 2 (related to 2H, 2I) can flow plots of the IL-5 versus IL-13 gated on either CD90.1+CD45.2+ or CD90.2+CD45.2+ ILC2 be shown? I.e. gate on the ILC2s and show cytokine expression, rather than the proportion of donor IL5/13. The proportion of donor ILC2 is shown to be significantly higher in 2G. Therefore gating on the cells of interest and showing on a cellular basis their ability to produce the cytokines would better make the point I think.

We agree that this is important additional data to include. We have added flow plots of sdLN ILC2s from the BM chimera divided by donor genotype showing IL-5 and IL-13 expression in New Figure S4H.

I assume the authors have looked and there is no obvious data, but does analysis of transcription factor consensus binding sequences in the open chromatin provide any new insight?

The Reviewer also commented on this in the public review. As copied from our response above:

We found that the most enriched sites in the ILC2 gene loci contained the consensus sequence GGGCGG (or its reverse complement), a motif recognized by a variety of zinc finger transcription factors (TFs). Predictions from our analyses predicted the KLF family of zinc finger TFs as most likely to be enriched at the identified open chromatin regions. To infer which KLFs might be occupying these sites in the RAG-experienced or RAG-naïve cells, we also assessed the expression levels of these identified TFs. Interestingly, KLF2 and KLF6 are more expressed in RAG-experienced ILC2s. KLF6 is a tumor suppressor (PMID: 11752579), and both KLF6 and KLF2 were recently shown to be markers of “quiescent-like” ILCs (PMID: 33536623). Further, upon analysis of the Th2 locus, the (A/T)GATA(A/G) consensus site (or reverse complement) was enriched in identified open chromatin at that locus. The algorithm predicted multiple TFs from the GATA family as possible binding partners, but expression analysis showed only GATA3 was highly expressed in ILC2s, consistent with what would be predicted from prior studies (PMID: 9160750).

We have added this data in new Figure S10 and new Figure S12, with corresponding text in the Results section on lines 277-316 and lines 366-378.

In terms of phrasing and presentation:- It would help to provide some explanation of why all analyses focus on the draining LNs rather than the actual site of inflammation (the ear skin). I do not think it appropriate to ask for data on this as this would require extensive further experimentation, but there should be some discussion on this topic. This feels relevant given that the skin is the site of inflammatory insult and ILC2 is present here. How the ILC2 compartment in the skindraining lymph nodes relates to those in the skin is not completely clear, particularly given the prevailing dogma that ILC2 are tissue-resident.

Given limitations of assessing cytokine production of the relatively rare population of skin-resident ILC2s, we focused on the skin-draining lymph nodes (sdLN). Our findings in the current manuscript are consistent with our prior work in Kim et al, 2013 (PMID 23363980), and more recently in Tamari et al, 2024 (PMID 38134932), which demonstrated correlation of increased ILC2s in sdLN with increased skin inflammation in the MC903 model. Similarly, Dutton et al (PMID 31152090) have demonstrated expansion of the sdLN ILC2 pool in response to MC903-induced AD-like inflammation in mice. We elaborate on the implications of our work for future studies, including limitations of our study (including the focus on the sdLN), and currently available reagents and need for new mouse strains to rigorously answer these questions on lines 476-508

- I think the authors should explicitly state that cytokine production is assessed after ex vivo restimulation (e.g. Lines 112-113).

We have added this statement to the revised text.

- I also think that it would help to be consistent with axis scales where analyses are comparable (e.g. Figure 1D vs Figure 1H).

We agree with the Reviewer and we have adjusted the axes for consistency. The data remains unchanged, but axes are slightly adjusted in New Figure 1 (D&I, E&J, F&K) and New Figure S2 (C-E match New Figure 1 D-F). This same axis scaling scheme is carried forward to New Figure 2 (D-E) and New Figure S4 (G,K,L). New data on cell counts is also included per request by Reviewers 2 and 3 (see above). However, we found results for total cells, including ILC2s (New Figure 1C,H, New Figure S2B, New Figure 2C, New Figure S4F), were consistent within experiments, but not between experiments, likely representing issues with normalizing counts (we did not include counting beads for more accurate total counts). Thus, the y-axes in those panels are not consistent between experiments/figures.

We feel reporting the proportion of WT vs *Rag1-/-* donor cells for the BM chimera is most illustrative of the effect of RAG and have kept it in the main New Figure 2, but for the BM chimera experiment panels we also include the total counts of IL-5^+^ and IL-13^+^ ILC2s (New Figure S4I,J).